# Structural basis of protein translocation by the Vps4-Vta1 AAA ATPase

Nicole Monroe, Han Han, Peter S Shen*, Wesley I Sundquist*, Christopher P Hill*

Department of Biochemistry, University of Utah School of Medicine, Salt Lake City, United States

**Abstract** Many important cellular membrane fission reactions are driven by ESCRT pathways, which culminate in disassembly of ESCRT-III polymers by the AAA ATPase Vps4. We report a 4.3 Å resolution cryo-EM structure of the active Vps4 hexamer with its cofactor Vta1, ADP·BeF$_x$, and an ESCRT-III substrate peptide. Four Vps4 subunits form a helix whose interfaces are consistent with ATP binding, is stabilized by Vta1, and binds the substrate peptide. The fifth subunit approximately continues this helix but appears to be dissociating. The final Vps4 subunit completes a notched-washer configuration as if transitioning between the ends of the helix. We propose that ATP binding propagates growth at one end of the helix while hydrolysis promotes disassembly at the other end, so that Vps4 'walks' along ESCRT-III until it encounters the ordered N-terminal domain to destabilize the ESCRT-III lattice. This model may be generally applicable to other protein-translocating AAA ATPases.

## Introduction

The AAA ATPase Vps4 drives the ESCRT (*Endosomal Sorting Complexes Required for Transport*) pathways that mediate membrane deformation and fission in a wide range of cellular processes (*Monroe and Hill, 2016*). These include membrane severing during cytokinetic abscission, the formation of multivesicular bodies and exosomes, shedding of microvesicles and viruses, repair of lesions in the plasma membrane, pruning of neurons, removal of defective nuclear pore complex assembly intermediates, and nuclear envelope closure at mitotic exit (*Campsteijn et al., 2016*; *Hurley, 2015*; *McCullough et al., 2013*; *Olmos and Carlton, 2016*). The ESCRT pathways converge on the recruitment of ESCRT-III subunits, of which seven family members are recognized in yeast and 12 in human. The leading model is that the upstream factors recruit ESCRT-III subunits, which polymerize through their N-terminal domains to induce an inherently unstable membrane configuration that resolves by fission following Vps4-mediated disassembly or remodeling of the ESCRT-III polymer (*Henne et al., 2013*; *McCullough et al., 2013*; *Schöneberg et al., 2017*).

Multiple structures have been reported for domains of Vps4 (*Monroe and Hill, 2016*). The N-terminal MIT domain of Vps4 (*Scott et al., 2005b*) binds ~20 residue MIT interacting motifs (MIMs) that are found at the C-termini of many ESCRT-III subunits (*Kieffer et al., 2008*; *Obita et al., 2007*; *Stuchell-Brereton et al., 2007*). The MIT domain is followed by a flexible ~40 residue linker and an ~320 residue AAA ATPase cassette that comprises a large AAA ATPase domain and a small AAA ATPase domain (*Figures 1A* and *2A*), which contains an insertion known as the β domain that binds the dimeric C-terminal VSL domain of the Vta1 cofactor (LIP5 in human) in an interaction that promotes Vps4 assembly and ATPase activity (*Azmi et al., 2006*; *Lottridge et al., 2006*; *Scott et al., 2005a*).

Vps4 is monomeric or dimeric in the cytosol, and assembles to form an active hexamer upon concentration at the membrane/ESCRT-III surface (*Monroe et al., 2014*). The central pore of this hexamer is thought to be lined by pore loop 1 and pore loop 2 (residues 203–210 and 240–248,

*For correspondence: peter.
shen@biochem.utah.edu (PSS);
wes@biochem.utah.edu (WIS);
chris@biochem.utah.edu (CPH)

Competing interest: See
page 18

Reviewing editor: Sriram
Subramaniam, National Cancer
Institute, United States

**eLife digest** Membranes surround multiple compartments within cells as well as the cell itself. In living cells, these membranes are remodeled continuously. This allows cells to divide, move molecules between different compartments and perform other essential activities. One important remodeling event is known as fission, which splits a membrane into separate parts.

Large repeating structures (or polymers) of ESCRT-III proteins play a crucial role in membrane fission. Breaking apart ESCRT-III polymers triggers membrane fission and also recycles the ESCRT-III proteins so that they can be used again.

An enzyme called Vps4 converts chemical energy (stored in the form of a molecule called ATP) into the mechanical force that breaks apart the ESCRT-III polymers. The active form of Vps4 consists of six Vps4 subunits working together to form a complex that includes a cofactor protein called Vta1. Monroe et al. have now used a technique called cryo-electron microscopy to determine the structure of an active yeast Vps4-Vta1 complex while it is bound to a segment of an ESCRT-III protein. This revealed that four of the six Vps4 subunits form a helix (which resembles a spiral staircase) that binds ESCRT-III in its central pore.

The structure implies that binding of ATP causes the Vps4 helix to grow at one end and that converting ATP into a molecule called ADP (to release energy) causes disassembly at the other end. The two additional Vps4 subunits move from the disassembling end to the growing end of the helix. In this manner, Vps4 'walks' along ESCRT-III, thereby pulling it through the pore at the center of the Vps4 complex and triggering breakdown of the ESCRT-III polymer. Further work is now needed to understand exactly how this activity leads to membrane fission.

respectively), which are highly conserved in AAA ATPases that act on protein substrates and play a critical role in substrate translocation (*Gonciarz et al., 2008*; *Han et al., 2015*; *Kieffer et al., 2008*; *Monroe and Hill, 2016*; *Scott et al., 2005a*). Importantly, nucleotide-induced asymmetry is required for binding of a peptide from an ESCRT-III subunit to the central pore of the Vps4 hexamer, and the 1:1 stoichiometry of this interaction (1 peptide to 1 hexamer) further confirms the asymmetric nature of the functional Vps4 complex (*Han et al., 2015*). Although considerable effort has been devoted to visualizing this active state, several structures of Vps4 assembled in various configurations (*Caillat et al., 2015*; *Hartmann et al., 2008*; *Landsberg et al., 2009*; *Yu et al., 2008*) have not provided a mechanistic model because they were determined for an inactive mutant, in the apo state, or in the presence of an inappropriate nucleotide.

Guided by the insights that Vps4 is active as a hexamer (*Monroe et al., 2014*) that is stabilized by binding of ADP·BeF$_x$ (*Han et al., 2015*) and the VSL domain of the Vta1 cofactor (*Azmi et al., 2006*; *Scott et al., 2005a*), we used *S. cerevisiae* proteins to prepare a Vps4-Vta1[VSL]-ESCRT-III[peptide]-ADP·BeF$_x$ complex for structural studies. Determination of this structure by cryo-EM revealed a highly asymmetric configuration in which four of the six Vps4 subunits form a helix that is stabilized by ATP and Vta1 binding, and is fashioned to bind substrate peptide in a $\beta$-strand conformation approximately along the helix axis. The structure implies a helix propagation mechanism in which binding of ATP promotes growth at one end of the Vps4 helix and ATP hydrolysis promotes disassembly at the other end, such that the Vps4 hexamer 'walks' along the ESCRT-III polypeptide, thereby conveying the ESCRT-III substrate through the central Vps4 pore in an extended conformation.

## Results and discussion

### Formation of a stable Vps4 hexamer complex

The yeast Vps4 construct used in these studies (residues 101–437) spans the AAA ATPase cassette. The active hexameric assembly was stabilized by expressing Vps4[101-437] with a C-terminal 18-residue linker followed by the hexameric *Pseudomonas aeruginosa* Hcp1 protein (*Mougous et al., 2006*). The active Vps4 conformation in the Hcp1 fusion protein was further stabilized by binding with ADP·BeF$_x$, an ESCRT-III peptide, and the Vta1[VSL] domain. Importantly, this fusion protein binds an

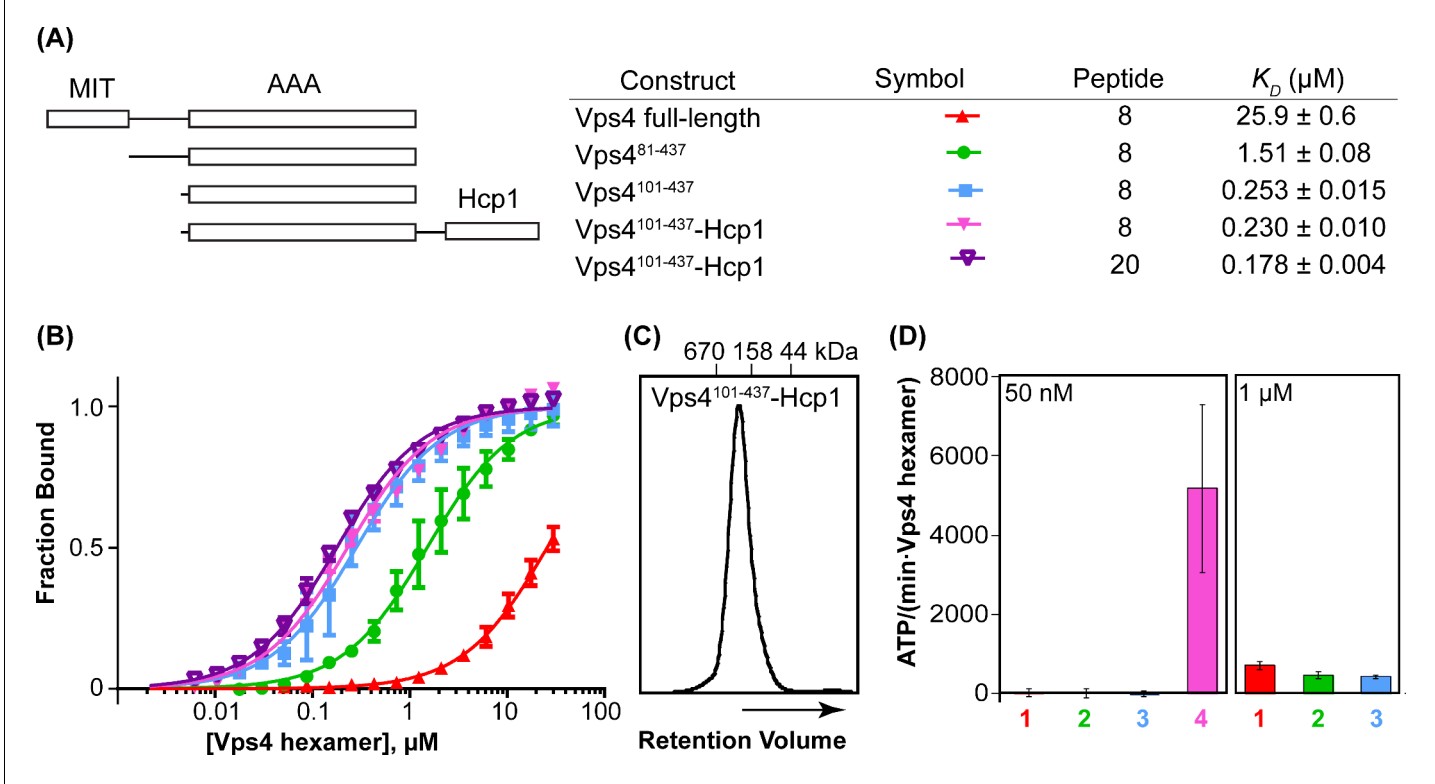

Figure 1. Vps4$^{101-437}$-Hcp1 is an active hexamer. (A) Vps4 constructs and peptide-binding affinities assayed by fluorescence polarization. Peptide '20' is a Vps2-derived 20-residue peptide C identified earlier (*Han et al., 2015*). Peptide '8' is an 8-residue fragment (DEIVNKVL) of peptide '20' that retains essentially full binding affinity. The relatively weak binding of full-length Vps4 reflects autoinhibition mediated by the MIT domains (*Han et al., 2015*). (B) Fluorescence polarization isotherms corresponding to values in panel A. Means and standard deviations are from four independent experiments. (C) Gel filtration of Vps4$^{101-437}$-Hcp1 on a Superdex 200 column in 25 mM Tris/HCl pH 7.4, 100 mM NaCl and 1 mM DTT. The protein elutes as a symmetric peak with an apparent molecular mass of 290 kDa, in good agreement with the calculated molecular mass of a hexamer (330 kDa). (D) ATPase activities for Vps4 constructs: 1, Vps4 full-length; 2, Vps4$^{81-437}$; 3, Vps4$^{101-437}$; 4, Vps4$^{101-437}$-Hcp1. Vps4 subunit concentrations are indicated. Means and standard deviations from at least three independent measurements.

The following source data and figure supplement are available for figure 1:

**Source data 1.** Binding of fluorescently labeled ESCRT-III peptides to Vps4, related to *Figure 1B*.

**Source data 2.** ATPase activity of Vps4 constructs, related to *Figure 1D*.

**Figure supplement 1.** Vps4$^{101-437}$-Hcp1 is a hexamer.

ESCRT-III substrate peptide with the same affinity as the Vps4 AAA ATPase cassette alone (*Figure 1AB*). The 8-residue ESCRT-III peptide used in these studies was derived from the ESCRT-III subunit Vps2 (residues 165–172), and binds Vps4 with essentially the same ~200 nM $K_D$ as the 20-residue parent peptide that we characterized in an earlier study (*Han et al., 2015*). As further controls, the fusion protein was found to elute from a size exclusion column as a single, symmetric peak (*Figure 1C*), to form a stable hexamer as shown by equilibrium sedimentation (*Figure 1—figure supplement 1*), and to be a highly active ATPase (*Figure 1D*). As shown below, other factors that indicate that Vps4 is not distorted by the Hcp1 fusion include the observation that Hcp1 has not imposed its 6-fold symmetry on the asymmetric Vps4 structure, the lack of contacts between Vps4 and Hcp1 in the overall consensus structure, and the short distance between Vps4 C-termini and Hcp1 N-termini (21–31 Å) compared to the 60 Å that could be accommodated by the fully extended 18-residue linker sequence.

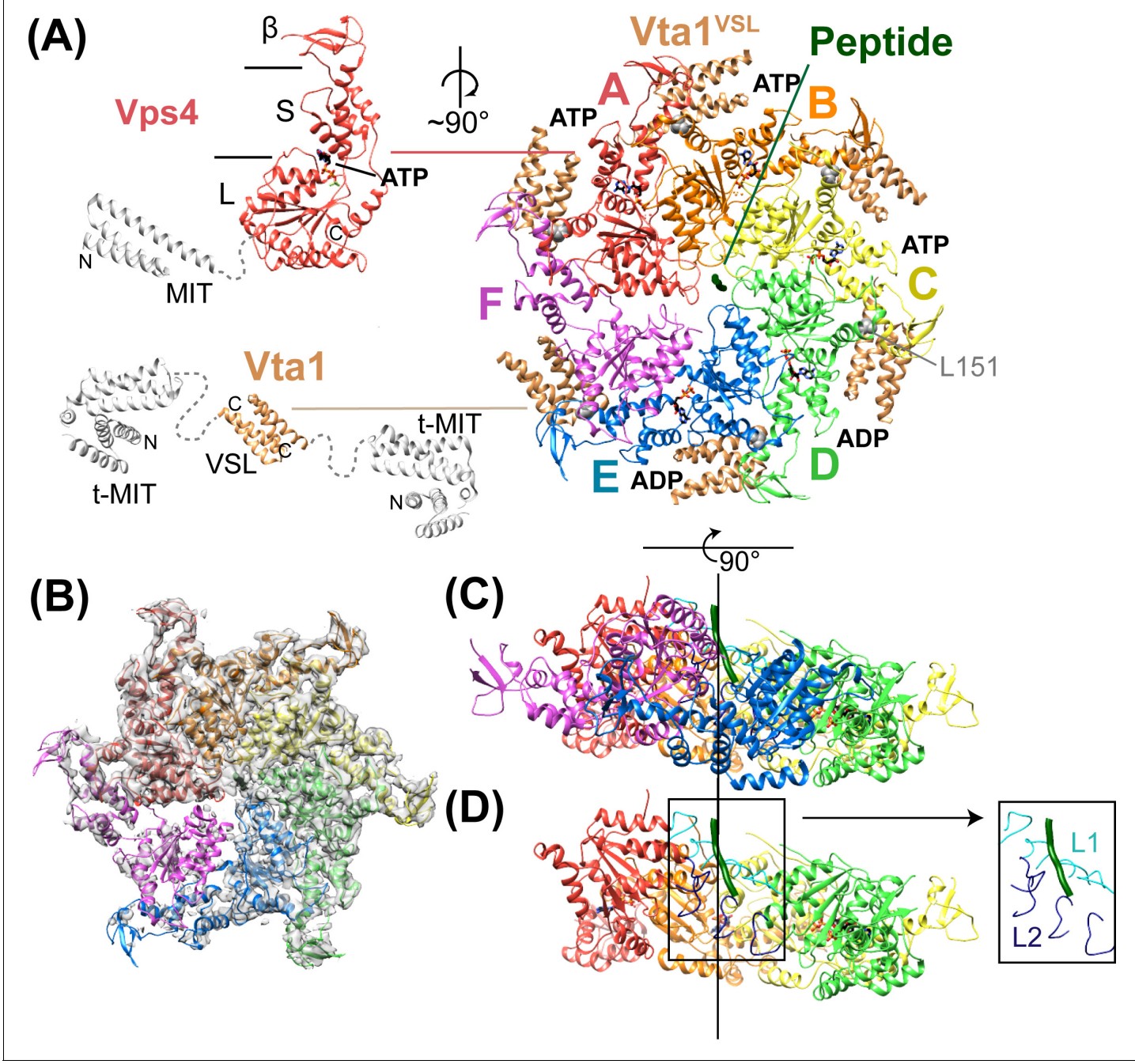

**Figure 2.** Structure of Vps4$^{101-437}$:Vta1$^{VSL}$:ESCRT-III$^{peptide}$:ADP·BeF$_x$. (**A**) Structure of the complex. The Vps4 and Vta1 constructs used for cryo-EM structure determination are shown in color on the left, with excluded segments colored white. MIT, large AAA ATPase (L), small AAA ATPase (S) and $\beta$ domains of Vps4 are labeled, as are the t-MIT and VSL domains of the Vta1 dimer. L151, a residue critical for hexamerization, is shown in gray spheres. (**B**) 4.3 Å map with the Vps4 model. (**C**) Side view of Vps4 hexamer, oriented with the subunit A-D helix axis vertical (black line). (**D**) Same as panel D but with subunits E and F removed. The inset shows the position of pore loops 1 (L1, residues 203–210, cyan) and pore loops 2 (L2, 240–248, dark blue) relative to the ESCRT-III peptide (dark green).

The following figure supplements are available for figure 2:

**Figure supplement 1.** Vps4 3D reconstruction, refinement, and validation.

**Figure supplement 2.** 3D reconstruction workflow.

*Figure 2 continued on next page*

*Figure 2 continued*

**Figure supplement 3.** Additional validation of the 3D reconstruction.
**Figure supplement 4.** Glutaraldehyde crosslinking improves the Vps4 density without distorting the structure.
**Figure supplement 5.** Refined model and representative density.
**Figure supplement 6.** Identification and classification of Vta1 density.
**Figure supplement 7.** Classification of subunit F Density.
**Figure supplement 8.** Rigid-body fitting of Vps4 subunit F.

## Structure determination

The Vps4 complex structure was determined by cryo-EM at 4.3 Å overall resolution to reveal a highly asymmetric hexameric ring of Vps4 subunits that bind the ESCRT-III peptide in the central pore and six Vta1$^{VSL}$ dimers around the periphery (*Figure 2*, *Table 1*, *Figure 2—figure supplements 1–6*). The local resolution varies from 4.0 to 5.0 Å over much of the AAA ATPase cassettes of Vps4 subunits A-E, and to 7 Å or lower resolution at the $\beta$ domains (*Figure 2—figure supplement 1*), Vta1$^{VSL}$ domains (*Figure 2—figure supplement 6*), and subunit F, which is distributed over at least three similar but distinct positions (*Figure 2—figure supplements 7–8*). The six Vps4 subunits adopt closely similar conformations but differ in the way that they contact each other. Although the limited resolution precludes detailed fitting of ADP·BeF$_x$, the structure implies that the distinct Vps4 interfaces are coupled to binding of ATP, hydrolysis to ADP·P$_i$, and nucleotide exchange. Importantly, the

**Table 1.** Reconstruction, refinement, and model statistics of Vps4.

|  | Vps4$^{101-437}$-Hcp1, whole particle | Hcp1-subtracted Vps4 |
| --- | --- | --- |
| **Reconstruction** | | |
| Particle images | 58,155 | 39,417 |
| Resolution (unmasked, Å) | 6.7 | 5.7 |
| Resolution (masked, Å) | 5.2 | 4.3 |
| Map sharpening B-factor (Å$^2$) | - | −208 |
| EM Databank Accession Number | EMD-8551 | EMD-8550 |
| **Refinement and validation of Vps4 subunits A-E** | | |
| Resolution used for refinement (Å) | - | 4.3 |
| Number of atoms | - | 10604 |
| R.M.S deviation | | |
| Bond length (Å) | - | 0.01 |
| Bond angles (°) | - | 0.92 |
| Ramachandran | | |
| Favored (%) | - | 91.13 |
| Allowed (%) | - | 8.87 |
| Outlier (%) | - | 0.00 |
| Molprobity score / percentile (%) | - | 1.94/100$^{th}$ |
| Clashscore / percentile (%) | - | 7.75/97$^{th}$ |
| PDB | - | 5UIE |

ATP and ADP·P$_i$ states can both be mimicked by ADP·BeF$_x$. We propose that the distinct nucleotide states progress sequentially around the hexameric ring (clockwise in *Figure 2A*), and that their stepwise conversion drives translocation of the ESCRT-III peptide, as discussed below.

## Overall structure

Vps4 subunits A-D form a right-handed helix that is created by the three very similar interfaces formed by the A-B, B-C, and C-D subunit pairs (*Figure 2*). These interfaces each bury ~2000 Å$^2$ of surface area and appear fashioned to coordinate ATP (*Wendler et al., 2012*), with R288 and R289 from the neighboring Vps4 subunit positioned to coordinate the ATP phosphates (*Figure 3*). The D-E interface (~1700 Å$^2$) is similar to the A-B, B-C, and C-D interfaces at the central pore region of the hexamer but deviates at the nucleotide-binding site, where the large AAA ATPase domain of subunit E is rotated by ~15° so that R288 and R289 are displaced by ~2 Å and are no longer able to coordinate see the nucleotide phosphates. We have modeled the nucleotide at this site as ADP, but are open to the possibilities that it may represent either ADP or ADP·P$_i$. The displacement of subunit E increases toward the hexamer periphery, and is further exaggerated by an ~10° increase in the hinge angle between the large and small AAA ATPase domains that allows the small AAA ATPase

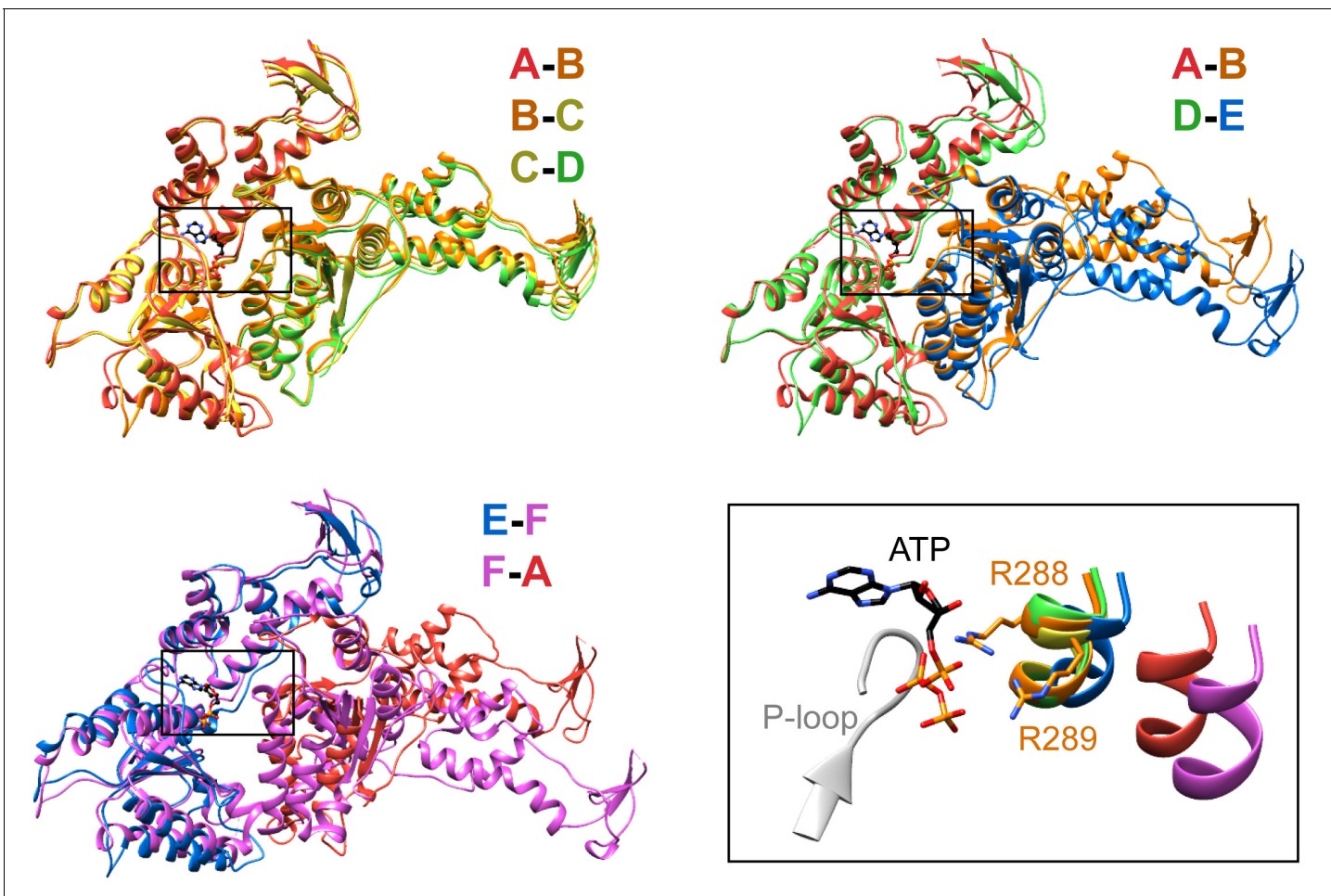

**Figure 3.** Interfaces in the asymmetric Vps4 hexamer. Vps4 subunit pairs superimposed on the large AAA ATPase domain of the first subunit, as indicated. A-B, B-C and C-D interfaces are equivalent. The nucleotide-binding site is slightly expanded at the D-E interface due to a 15° relative rotation of subunit E. The E-F and F-A sites are open for nucleotide exchange. Inset (black rectangle), Close-up on the nucleotide binding site showing the nucleotide and coordinating P-loop for the first subunit, with the R288/R289-containing helix of the second subunits in color. These arginine finger Cα atoms shift by 2 Å at the D-E interface relative to the A-B, B-C, and C-D subunits. The E-F and F-A interfaces are shifted by 8 Å and 16 Å, respectively.

domain of subunit E to maintain contact with subunit F. In contrast, the E-F and F-A interfaces, which bury only 500 and 900 Å$^2$, respectively, maintain contacts primarily near the hexamer periphery and appear open to allow nucleotide exchange.

## Vta1 dimers bind two adjacent Vps4 subunits

The Vta1$^{VSL}$ dimers, which are pairs of helical hairpins that form a 4-helix bundle, stabilize the ring by forming struts between adjacent Vps4 subunits (*Figure 4A*). Their density is clearly visible only after focused 3D classification (*Figure 2—figure supplement 6*), which reveals distinct density for all four helices of the Vta1$^{VSL}$ dimers adjacent to the β domains of Vps4 subunits A and B, while density for individual helices is less clearly defined but still apparent for Vta1$^{VSL}$ at subunits C and F (*Figure 4—figure supplement 1*). Subunits D and E do not show distinct VSL helices, but do show some overall density for the VSL bundle. This variation in the quality of Vta1$^{VSL}$ densities among the six interfaces likely reflects differences in occupancy and binding modes at each site, as discussed below.

In all cases, the Vta1$^{VSL}$ dimer contacts the β domain of one Vps4 subunit in an interface that we have modeled according to a previously reported crystal structure (*Yang and Hurley, 2010*) in which Y303 and Y310 of the first Vta1 subunit contact two loops of the β domain (residues 356–359 and 375–385) (*Figure 4B*). Compared to the earlier crystal structure, the Vta1$^{VSL}$ construct in our structure is extended by 10 residues at the N-terminus, including residues of the 'Vps4 stimulatory element' (*Norgan et al., 2013*). Some density for these residues is visible for the second subunit of the best-defined Vta1$^{VSL}$ dimers, where they make a small 70 Å$^2$ contact with α7 and α9 in the small AAA ATPase domain of the same Vps4 subunit, as suggested previously (*Davies et al., 2014*) (*Figure 4B*, *Figure 4—figure supplement 1C*). Consistent with their more clearly defined density, the Vta1$^{VSL}$ dimers bound to the β domains of subunits A, B, C, and F also contact the small AAA ATPase domain of the following Vps4 subunit (*Figure 4C*), with Y310' and surrounding residues at the hairpin end of the second Vta1 subunit contacting α6 and α7 (residues 300–330) of the small AAA ATPase domain of the second Vps4. This novel interaction shows an unusual use of the two-fold symmetry-related VSL dimer residues, Y310 and Y310', to contact different surfaces on neighboring Vps4 subunits, which is consistent with our biochemical finding that Vta1 stabilizes formation of the hexamer (*Monroe et al., 2014*; *Scott et al., 2005a*) rather than higher-order assemblies (*Xiao et al., 2008*; *Yang and Hurley, 2010*).

Vta1$^{VSL}$ dimers can bind in the same manner to the Vps4 subunit pairs F-A, A-B, B-C, C-D, but interactions at the D-E and E-F subunit pairs appear to be suboptimal. Superposition performed on the helices of the small AAA ATPase domains of the first Vps4 in the subunit pairs (*Figure 4D*), which are relatively well defined, shows that Vta1 can make superimposable interactions with both the β domain of the first Vps4 subunit and with the small AAA ATPase domain of the second Vps4 subunit for the A-B, B-C, and C-D subunit pairs, and that F-A is quite similar. In contrast, the D-E and E-F interfaces are incompatible with Vta1 forming the same contacts between neighboring Vps4 subunits as seen at the A-B, B-C, C-D, and F-A interfaces (*Figure 4D*). Instead, our preferred interpretation of the density is that Vta1$^{VSL}$ at the D-E and E-F interfaces remains bound to the β domain of the first Vps4 subunit but cannot form optimal contacts with its adjacent subunit.

To test the importance of the interface seen between Vta1 and the small AAA ATPase domain of the second subunit, we quantified binding of Vta1$^{VSL}$ to the Vps4$^{101-437}$-Hcp1 hexamer and to the Vps4$^{101-437}$ L151D mutant, which is predominantly monomeric in the absence of Vta1$^{VSL}$ (*Gonciarz et al., 2008*). The Vps4$^{101-437}$-Hcp1 hexamer showed an ~30 fold tighter apparent $K_D$ than Vps4$^{101-437}$ L151D (*Figure 5*), which supports our observation that Vta1$^{VSL}$ binds to two neighboring Vps4 subunits in the hexamer and is consistent with a role for Vta1 in stabilizing assembly of the active Vta1 hexamer (*Azmi et al., 2006*; *Scott et al., 2005a*). Residues K321, E322 and R325 of the small AAA ATPase domain of the second Vps4 subunit are located in the vicinity of residue Y310' of the VSL domain (*Figure 5A*). Consistent with their proximity to a region of the Vta1 surface that has negative electrostatic potential, mutating the lysine and arginine residues to alanine and aspartate, respectively, decreases binding to the Vps4 hexamer, whereas mutation of the glutamate to alanine increases binding affinity (*Figure 5B*).

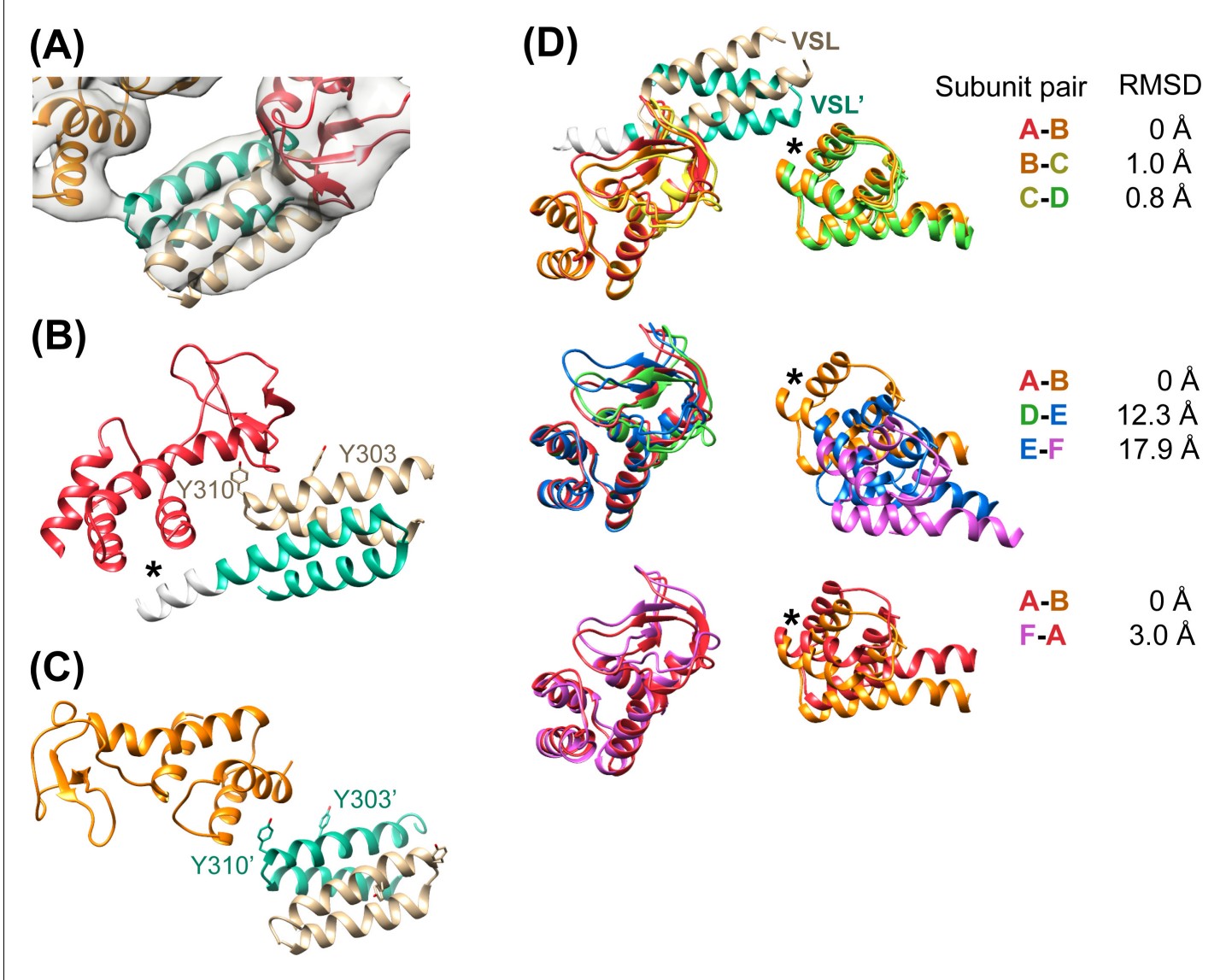

**Figure 4.** Vta1[VSL] contacts with Vps4. (A) Density for the most clearly defined Vta1[VSL] (bound to the Vps4 subunit A β domain). The Vta1[VSL] subunits are colored tan and teal. (B) Vta1[VSL] interaction with the first Vps4 subunit. This interface is modeled identically to a crystal structure of Vta1[VSL] in complex with a truncated Vps4 construct (*Yang and Hurley, 2010*). Additional N-terminal residues in the longer Vta1 construct used in this study are shown in white and their interaction with the small AAA ATPase domain of Vps4 is indicated with an asterisk. (C) Vta1[VSL] interaction with the second Vps4 subunit. Y303' and Y310' are labeled. (D) Overlap of subunit pairs on the small AAA ATPase domain of the first Vps4 (residues 301–349 and 403–411). Consequent RMSD values are shown for residues 300–311 and 320–331 of the second Vps4 subunit at the second Vta1 interface (asterisk).

The following figure supplement is available for figure 4:

**Figure supplement 1.** Rigid-body fitting of Vps4 β domain-Vta1[VSL] complexes into each density map.

## ESCRT-III substrate peptide binds close to the helix axis of the central pore

Density accommodates the 8-residue ESCRT-III peptide in an extended conformation. This density extends weakly at both ends and is not sufficiently clear to reliably build a model with side chains, consistent with the possibility that binding may occur in several overlapping positions (*Figure 6*). The primary contacts are with pore loop 1 (residues 203–209) of the A, B, C, and D Vps4 subunits, whose closest Cα atoms are ~7.5 Å from the helix axis that is defined by these loops. There is also a

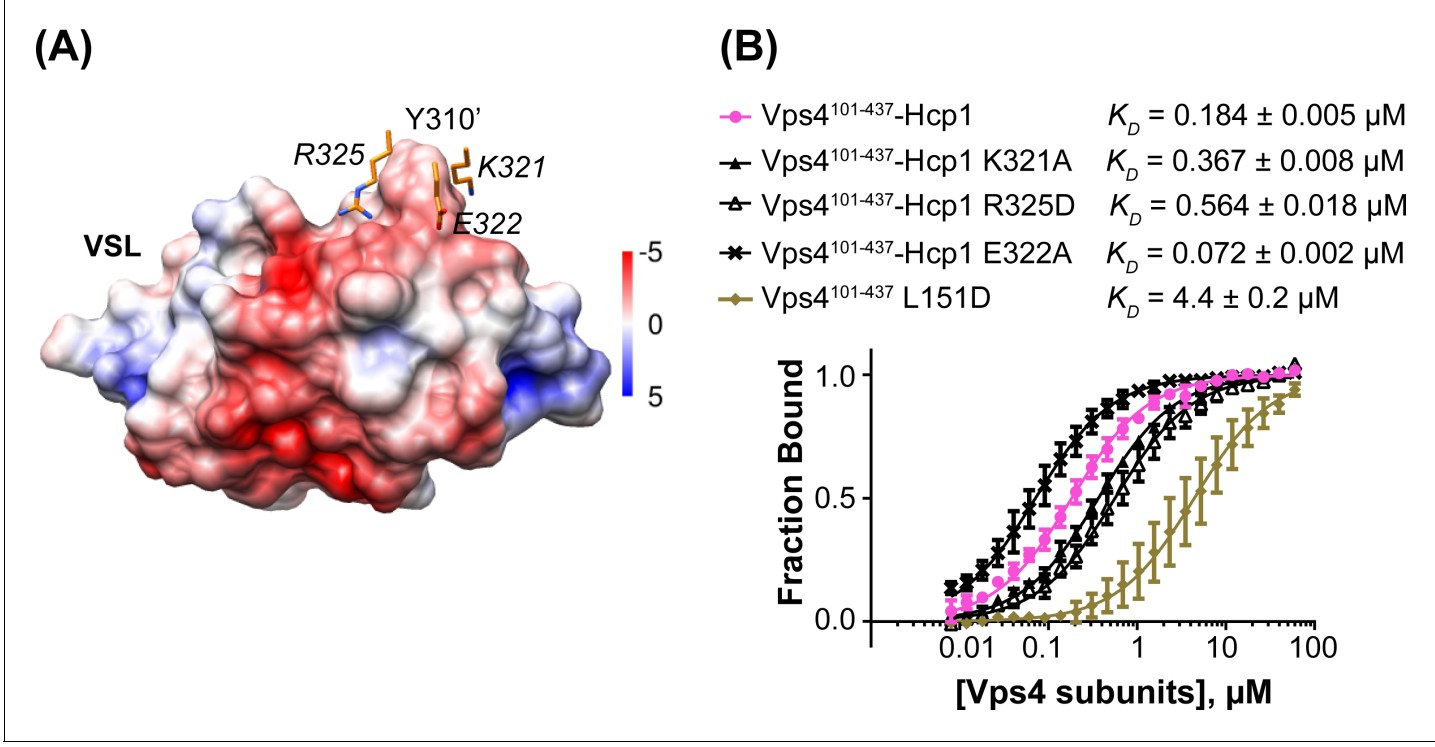

**Figure 5.** Mutations at the Vta1 interface with the second Vps4 subunit alter binding affinity. (A) K321, E322 and R325 of the Vps4 small AAA ATPase domain (orange, labels in italic font) contact Y310' of Vta1$^{VSL}$ (surface representation colored by electrostatic potential, kT/e) in the interaction shown in *Figure 4C*. (B) Binding of fluorescently labeled Vta1$^{VSL}$ to the Vps4$^{101-437}$-Hcp1 hexamer (pink circles) is ~24x tighter than binding to a monomeric Vps4 construct, Vps4$^{101-437}$ L151D (gold diamonds). Consistent with the Vta1 surface electrostatic potential, point mutations K321A and R325D weaken Vta1$^{VSL}$ binding 2-fold and 3-fold, respectively, while E322A strengthens binding 2-fold. Means and standard deviations are from at least three biological replicates.

The following source data is available for figure 5:

**Source data 1.** Binding of Vta1$^{VSL}$ to Vps4, related to *Figure 5B*.

contact with pore loop 1 of subunit E, although the loop density in that subunit is relatively weak. The peptide lies approximately along the helix axis, with Cα atoms modeled 1.0–2.7 Å (average 1.6 Å) from the axis, which is consistent with the model that substrates are translocated along or close to the helix axis, with some variation allowed to accommodate distinct amino acid sequences. The helical symmetry of pore loop 1 of Vps4 A-D is approximately continued by subunit E and has successive loops separated by a translation of 6.3 Å along the helix axis and a rotation of 60° (*Figure 7A*). This matches the translation and rotation seen every two residues along a canonical β-strand, such that successive dipeptides of a β-strand that lies approximately along the helical axis could make equivalent interactions with pore loop 1 residues of successive Vps4 subunits. Hence, these four loops present a curved peptide-binding surface that extends into the hexamer pore. Pore loop 1 of subunit F is displaced ~14 Å from the helix axis and is therefore completely disengaged from the substrate (*Figure 7A*). Although we do not observe a contact between the peptide density and residues of Vps4 pore loop 2 (residues 241–251) and the density of pore loop 2 is generally quite poor, we note that these loops of the Vps4 A-D subunits are arrayed contiguously with the pore loop 1 residues through the hexamer pore and follow the same helical symmetry, which is consistent with the possibility that they continue the substrate binding surface used by Vps4 to translocate ESCRT-III subunits.

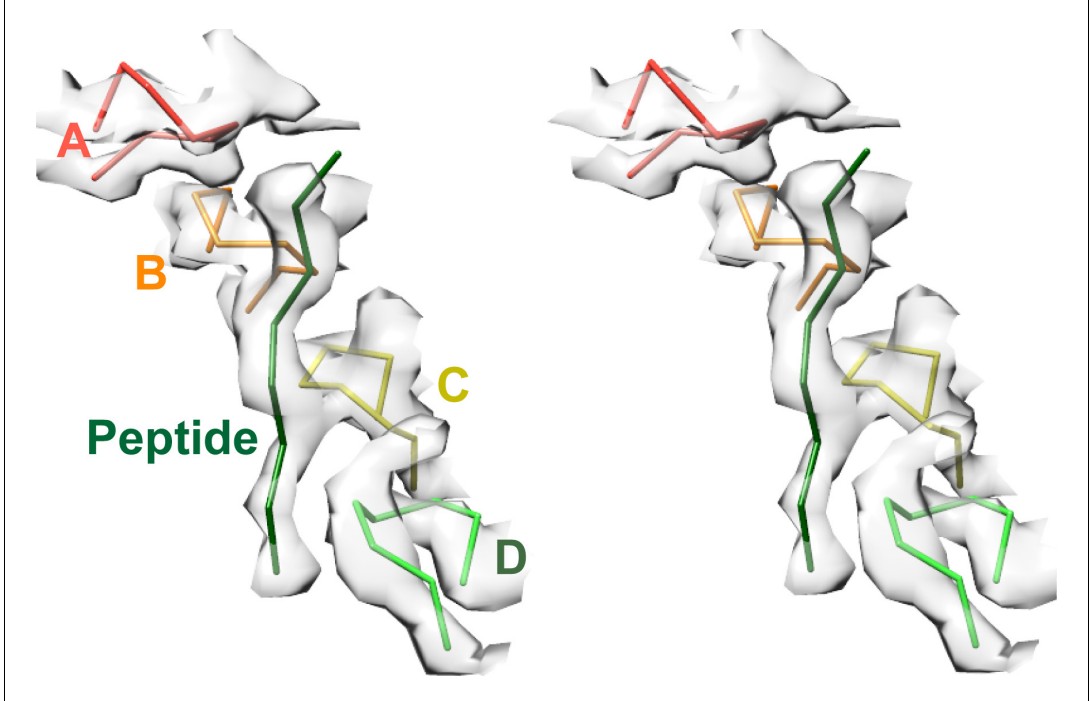

**Figure 6.** Pore loops of Vps4 form a spiral staircase to coordinate the substrate peptide. Stereo view of the peptide and pore loop 1 (residues 203–210) of subunits A-D with density.

## Helix propagation mechanism of substrate translocation

We propose that Vps4 translocates its ESCRT-III substrates through the hexamer pore by a helix propagation mechanism in which binding of ATP and Vta1 promotes growth at the A-end of the Vps4 helix while ATP hydrolysis and release promotes dissociation at the D/E-end of the helix (*Figure 7B*, *Figure 8*, *Videos 1* and *2*). In repeated cycles, the propagating 4-subunit Vps4 helix will 'walk' along the substrate while binding it in an extended β-strand conformation and conveying it through the hexamer pore. Some variation in strand conformation may be tolerated but the overall effect is expected to be unfolding of ESCRT-III structure as the Vps4 helix advances. The structure indicates that Vta1 will promote this process by binding adjacent Vps4 subunits in the helical conformation. Moreover, as seen in the structure, Vta1[VSL] forms the same helix-promoting interaction between subunits F and A, as if pulling subunit F into an ATP-binding position at the leading end of the helix. At the other end of the helix, ATP hydrolysis correlates with expansion of the D-E nucleotide site to trigger disassembly of the helix, disengagement from the substrate by subunit F, opening of the interface to allow nucleotide exchange, and subsequent rebinding at the leading end of the propagating helix. An attractive feature of this model is that the symmetry match between a β-strand ESCRT-III substrate and the pore loop 1 residues of the Vps4 helix means that each of the Vps4 subunits can make identical interactions with consecutive ESCRT-III dipeptides, and that these interactions do not need to change during the translocation process.

## A role for avidity and hexamer formation in Vps4 function

The inherently weak hexamerization of Vps4 (*Monroe et al., 2014*) likely contributes to substrate specificity by coupling Vps4 recruitment to assembly of the active hexamer. As illustrated in *Video 3*, we envision that Vps4 concentrates at ESCRT-III polymers by binding of its MIT domain with MIMs at the ESCRT-III C-termini. Vta1 possesses an N-terminal tandem MIT domain (t-MIT) (*Xiao et al., 2008*) that can also bind MIM sequences and is connected to the VSL domain by a flexible ~100 residue linker. Because Vps4 and Vta1 also bind to each other, the concentrating effect at ESCRT-III polymers will be synergistic, and will promote Vps4 binding to ATP and hexamerization around the 30–170 flexible residues that lie between the MIM and the folded N-terminal domain of various ESCRT-

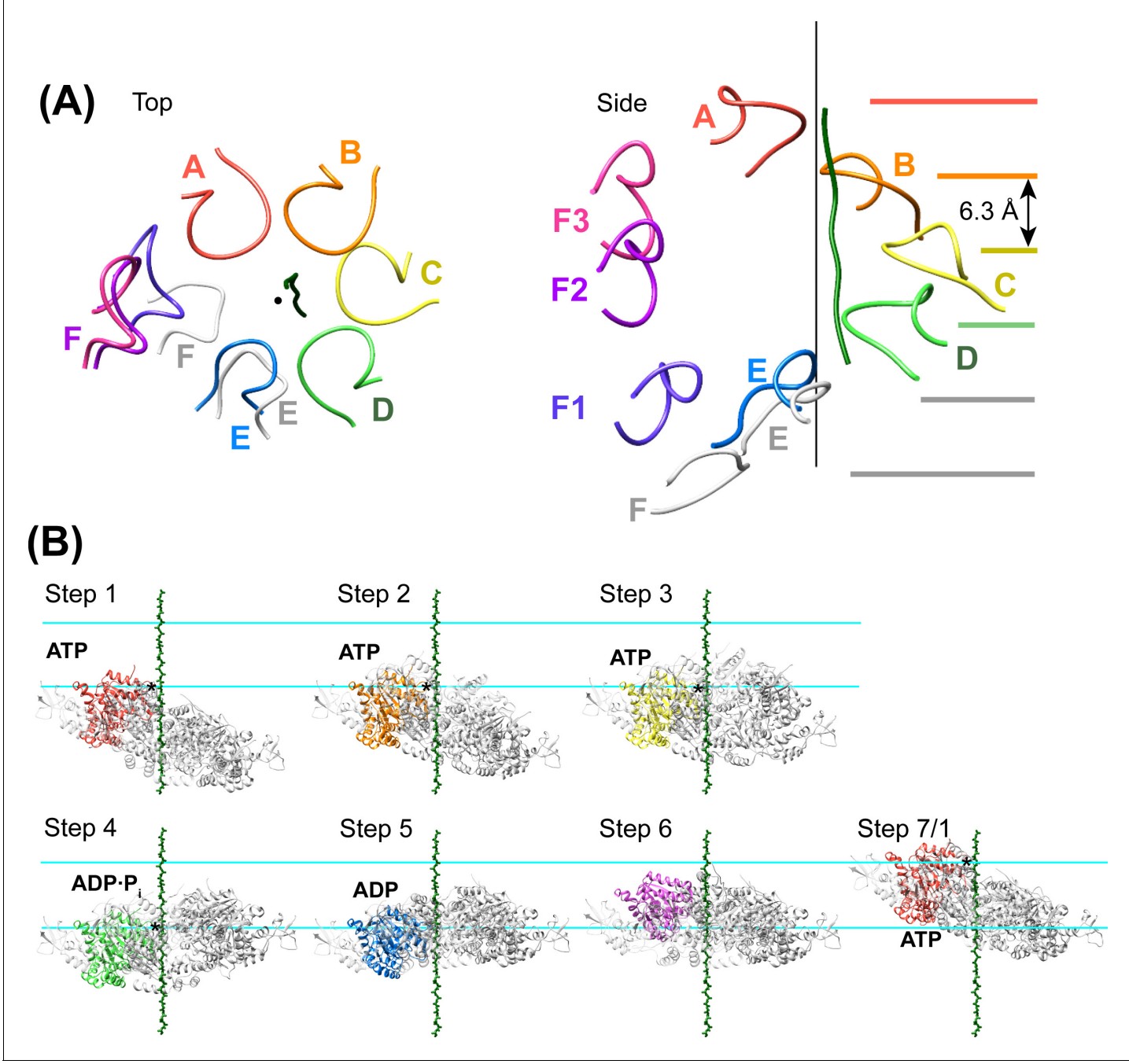

**Figure 7.** Peptide binding and mechanism of translocation. (**A**) The pore loop 1 residues of subunits A-D form a helix (axis, black line) that matches the symmetry of a canonical twisted $\beta$-strand, which rotates 60° and translates 6.3 Å every two residues. In white are the positions that subunits E and F would adopt if they continued this helix. The three positions seen for subunit F (*Figure 2—figure supplements 7–8*) appear to be snapshots along the return path from the end of the helix at subunit E to the start of the helix at subunit A. (**B**) Steps along the translocation cycle inferred from the cryo-EM structure. The peptide shown is modeled as a $\beta$-strand along the helix axis of subunits A-D. Vps4 maintains a constant interaction with the peptide through steps 1 to 4 before dissociating at step 5 and rebinding 12 residues further up the peptide at step 7, which is equivalent to step 1. Nucleotides suggested by density and coordination geometry are labeled. Pore loop 1 contacts with the substrate peptide in steps 1–4 are indicated with an asterisk. The two subunits closest to the view direction are included with 50% transparency. The two horizontal lines are separated by 37.8 Å (12 residues) and indicate points of substrate contact with pore loop 1 of the highlighted subunit.

The following figure supplements are available for figure 7:

**Figure supplement 1.** Comparison of the Vps4 hexamer with the ATPases of the 26S proteasome.

*Figure 7 continued on next page*

*Figure 7 continued*

**Figure supplement 2.** Comparison of the Vps4 hexamer with the NSF D1 ring.

III subunits (*Han et al., 2015*). The Vps4 complex can subsequently hydrolyze ATP to translocate along the polypeptide until the ESCRT-III N-terminal domain is destabilized and removed from the polymer. If the polymer remains intact after removal of one subunit, Vps4 and Vta1 would remain at high concentration due to the continuing MIM-MIT interactions and so could reassemble around another available ESCRT-III C-terminal sequence to repeat the process for as long as the ESCRT-III polymer persists to present an array of MIMs.

## Unresolved mechanistic questions

The model that Vps4 translocates toward the N-terminal domain of ESCRT-III implies that substrate binds to the hexamer pore in a defined direction, which is a level of detail that is not resolved in our current structure. Also unresolved is the biological role of the 8-residue binding peptide that we identified from Vps2, which based upon structural information available for other ESCRT-III proteins likely corresponds to a short helix that packs against the folded core in an isolated ESCRT-III subunit (*Bajorek et al., 2009*; *Xiao et al., 2009*) and in an ESCRT-III polymer (*McCullough et al., 2015*). These structures suggest that the peptide sequence disengages from the folded ESCRT-III core to bind Vps4 and raise the possibility that Vps4 destabilizes ESCRT-III structure simply by binding to this sequence, although our preferred model is that Vps4 initially hexamerizes around a more C-terminal segment and subsequently translocates beyond the site of the 8-residue peptide such that the entire ESCRT-III subunit is destabilized (*Yang et al., 2015*).

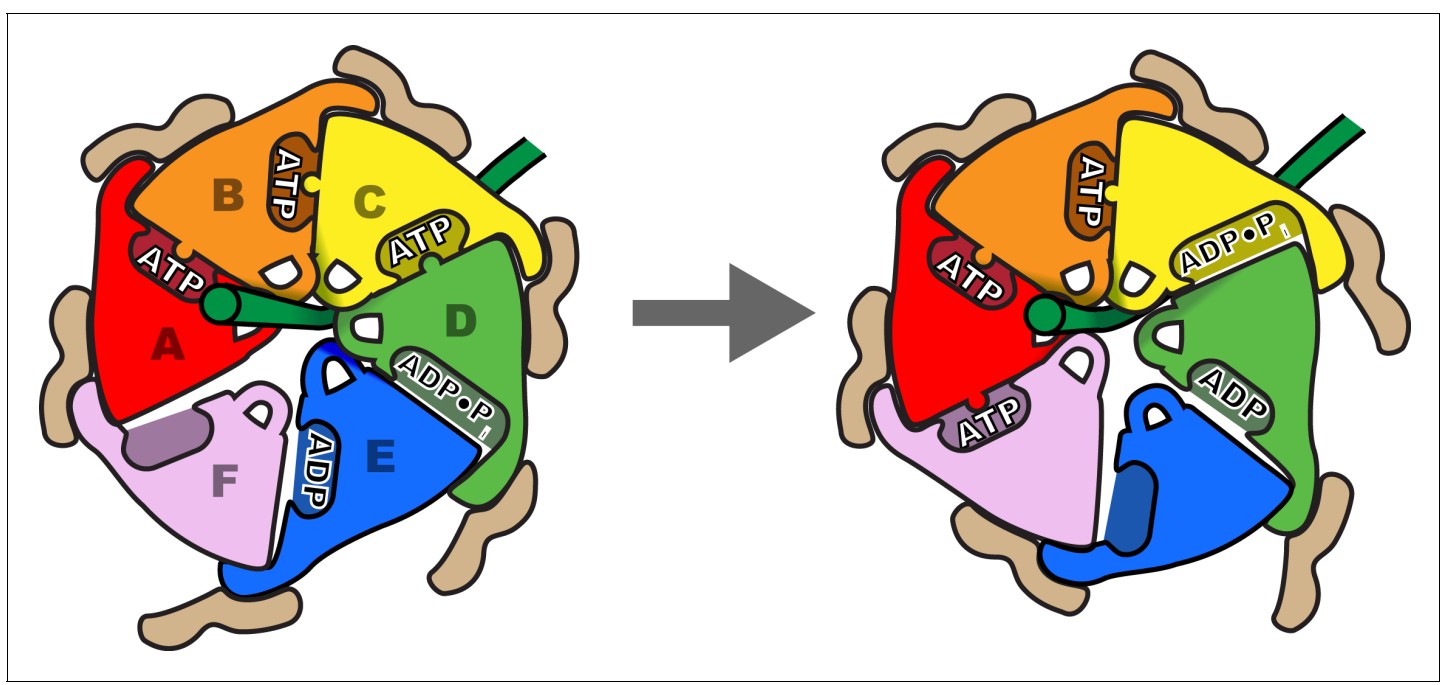

**Figure 8.** Schematic of one step in the translocation mechanism. *Left,* Subunits A-D form a helical surface of pore loop 1 residues that binds substrate in a $\beta$ conformation along or close to the helix axis. The helix is stabilized by Vta1$^{VSL}$ binding to adjacent subunits and by ATP binding at subunit interfaces. *Right,* next step in the cycle where subunit F has bound ATP and assembled on the growing end of the Vps4 helix, ATP has been hydrolysed at the C-D interface, and the nucleotide-binding site of subunit E has been opened to allow ADP·P$_i$ release and rebinding of ATP.

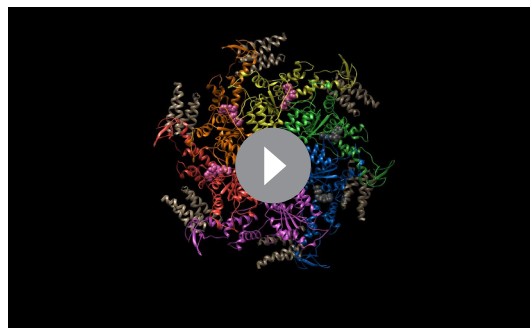

**Video 1.** Top view of the proposed translocation mechanism. Vps4 reaction cycle modeled by linear interpolation between the six different states represented in the cryo-EM structure. The ESCRT-III substrate is modeled as a $\beta$-strand lying along the axis of the helix defined by Vps4 subunits A-D. Nucleotides are shown in pink (ATP) and gray (ADP).

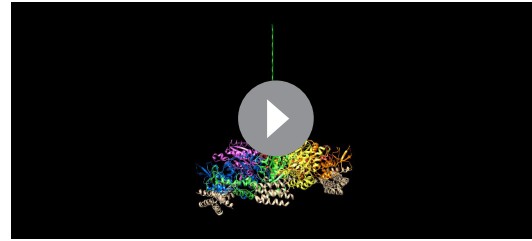

**Video 2.** Side view of the proposed translocation mechanism. As for *Video 1*.

## Comparison with other AAA ATPases

Structures of the substrate-bound DNA helicase E1 (*Enemark and Joshua-Tor, 2006*) and RNA translocase Rho (*Thomsen and Berger, 2009*; *Thomsen et al., 2016*) suggested sequential mechanisms of polynucleotide translocation that are conceptually analogous to our proposal for polypeptide translocation by Vps4. Thus, the structures indicate that Vps4 and the hexameric nucleic acid translocases function by forming a helical arrangement of subunits that matches the symmetry of their translocating substrate, with one or two transitioning subunits that are disengaged from the substrate, and with translocation achieved by sequential propagation of the ATPase helix.

The structure of Vps4 superimposes with other hexameric AAA ATPases, including the ATPases of the 26S proteasome, which, like Vps4, translocate protein substrates and adopt a right-handed helical notched-washer structure (*Förster et al., 2013*; *Huang et al., 2016*; *Lander et al., 2013*). Multiple proteasome structures show four or five ATPase subunits forming a helix in which the pore loop 1 residues overlap Vps4 with RMSD values of ~2.3 Å (*Figure 7—figure supplement 1*). The ATP-bound conformation of the AAA ATPase NSF (*Zhao et al., 2015*) shows a similar overlap of pore loops (*Figure 7—figure supplement 2*), although the extent to which it may translocate substrate is unclear. It will be of considerable interest to determine the extent to which the geometry of substrate binding and the cycles of ATP-induced helix propagation envisioned here in light of the Vps4-substrate complex underlie the mechanisms of other protein-translocating AAA ATPases, and the extent to which variations on the idealized model, such as by variations in peptide binding geometry or in the sequence and timing of ATP hydrolysis, may apply.

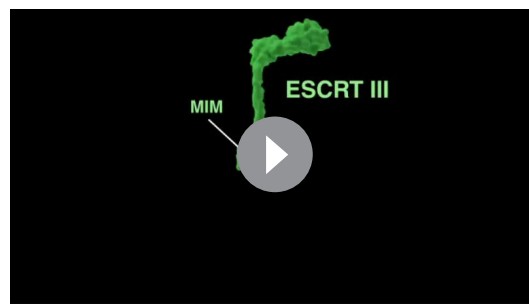

**Video 3.** Model of Vps4 assembly and ESCRT-III disassembly. Vps4 (purple) is recruited to the ESCRT-III lattice (green) through binding of its N-terminal MIT domain to *M*IT *I*nteracting *M*otifs (MIMs) (*Kieffer et al., 2008*; *Obita et al., 2007*; *Stuchell-Brereton et al., 2007*), which are sequences at the ends of the long, flexible C-terminal tails of ESCRT-III subunits. The avidity effect of the ESCRT-III polymer promotes Vps4 hexamerization around flexible ESCRT-III sequences. The hexamer is further stabilized by the dimeric Vta1 protein (tan), which also concentrates at the ESCRT-III polymer through its N-terminal t-MIT domain (not shown) (*Guo and Xu, 2015*; *Skalicky et al., 2012*; *Vild et al., 2015*). The Vps4 hexamer hydrolyzes ATP and translocates the substrate through the central pore, thereby destabilizing the ESCRT-III structure and removing the subunit from the lattice. We speculate that Vps4 and Vta1 remain bound to the ESCRT-III lattice via their MIT domains such that they are in position to process additional ESCRT-III subunits until the polymer is disassembled. This animation was created using Autodesk Maya (Autodesk, Inc.) from protein structural models exported from UCSF Chimera (*Pettersen et al., 2004*).

## Materials and methods

### Protein expression and purification

Vps4 and Vta1 proteins were expressed in *E. coli* BL21(DE3)RIL (Agilent Technologies, Santa Clara, CA) from a pET151-based vector with a cleavable N-terminal 6xHis tag. Proteins and expression vectors used in this study are listed in *Supplementary file 1*. Expression cultures were grown in ZY auto-induction media (*Studier, 2005*) at 37°C for 6 hr and at 19°C for 16 hr. Cells were harvested by centrifugation and stored at −80°C. The same purification strategy was used for all Vps4 and Vta1 constructs. Cell pellets were thawed and resuspended in lysis buffer (50 mM Tris/HCl pH 8.0, 300 mM NaCl, 5% (v/v) glycerol, 10 mM imidazole) supplemented with protease inhibitors, 1 mg of DNA-seI and 100 mg of lysozyme. After incubation on ice for 30 min, cells were lysed by sonication. Lysate was clarified by centrifugation and batch-bound to 10 ml of Ni-NTA agarose equilibrated in lysis buffer. Following a wash with 150 ml of lysis buffer, His-tagged protein was eluted with lysis buffer made up with 75 mM imidazole (500 mM for Vps4$^{101-437}$-Hcp1 fusions). His tags were cleaved by incubation with 1 mg of TEV protease during dialysis against 25 mM Tris/HCl pH 8.0, 150 mM NaCl, 1 mM EDTA, 1 mM DTT overnight at 4°C. Samples were then dialyzed into 25 mM Tris/HCl pH 8.0, 100 mM NaCl for 4 hr with two buffer changes and incubated with 10 ml of Ni-NTA agarose equilibrated with dialysis buffer to remove the cleaved His tags and the His-tagged TEV protease. The sample was subsequently bound to a 5 ml HiTrap Q FF ion exchange column (GE Healthcare) equilibrated with Q buffer A (25 mM Tris/HCl pH 8.0, 100 mM NaCl, 1 mM DTT), and eluted with a gradient of 0–50% Q buffer B (25 mM Tris/HCl pH 8.0, 1 M NaCl, 1 mM DTT) over 30 column volumes. Fractions containing the protein of interest were pooled, concentrated to ~5 ml and further purified by gel filtration into 25 mM Tris/HCl pH 7.4, 100 mM NaCl, 1 mM DTT or 20 mM HEPES/NaOH pH 7.4, 100 mM NaCl, 1 mM DTT using a 120 ml Superdex 200 column (GE Healthcare, Chicago, IL). The yield per liter of expression culture was typically 20–30 mg for Vps4 proteins and 40–50 mg for Vta1$^{VSL}$.

### Peptide synthesis

Peptides DEIVNKVL (Vps2 residues 165–172) and DEIVNKVLDEIGVDLNSQLQ (Vps2 residues 165–184) were prepared by solid-phase synthesis on a Prelude X peptide synthesizer (Protein Technologies, Inc., Tucson, AZ) using standard procedures and Fmoc chemistry (*Chan and White, 2000*). Unlabeled peptides were N-terminally acetylated and C-termini were produced as carboxyamides. For peptides used in fluorescence polarization assays, 5 (6)-carboxyfluorescein (Acros Organics, Geel, Belgium) was coupled to the N-terminal α-amine by standard coupling conditions. Following cleavage from resin by TFA, the peptides were precipitated with ice-cold ether, washed thoroughly with ether, and dried overnight under vacuum. Peptides were then HPLC purified on a Phenomenex 5 μm-C4 Jupiter column (10 × 250 mm, 300 Å) at 5 ml/min over a 15 min gradient (20–80% ACN,0.1% TFA). Peptide quality was verified by LC-MS using a FortisBIO 5 μm C4 column (4.6 × 150 mm, 300 Å) coupled to an Agilent 6100 Series Single Quadrupole mass spectrometer.

### Fluorescence polarization assay for ESCRT-III peptide binding

Binding of fluorescently labeled peptides to Vps4 was quantified in the presence of a 2:1 ratio of Vta1$^{VSL}$:Vps4 subunits and 1 mM ADP·BeF$_x$, as described previously (*Han et al., 2015*). Briefly, peptides (1 nM) were incubated at room temperature with Vps4:Vta1 complexes (0–180 μM Vps4 subunits) in binding buffer (20 mM HEPES/NaOH, pH 7.4, 100 mM NaCl, 1 mM ADP·BeF$_x$, 10 mM magnesium chloride, 1 mM DTT) in a total volume of 60 μl. When equilibrium was reached, parallel and perpendicular fluorescence intensities were measured on a Biotek Synergy Neo HTS microplate reader using an excitation wavelength of 485 nm and an emission wavelength of 528 nm. Because Vps4 binds a single peptide per hexamer (*Han et al., 2015*), fluorescence polarization was plotted against the Vps4 hexamer concentration and dissociation constants were estimated by global fitting of the equation FP = [Vps4 hexamer]/($K_D$ + [Vps4 hexamer]) to data points from four independent experiment, where FP is the normalized fluorescence polarization or 'fraction bound' and independent experiments are defined as using different protein preparations, using GraphPad Prism 6 (GraphPad Software, Inc., La Jolla, CA).

## Fluorescence polarization assay for Vta1[VSL] binding

For binding studies, Vta1[VSL] S278 was replaced by cysteine. Fluorescent modification of the single cysteine was performed by incubating 8 µM Vta1[VSL] S278C with 200 µM fluorescein-5-maleimide (Fisher Scientific) in 25 mM Tris/HCl pH 7.4, 100 mM NaCl, 5 mM EDTA overnight at 4°C. Excess label was removed using a PD-10 desalting column equilibrated in binding buffer (20 mM HEPES/NaOH pH 7.5, 100 mM NaCl, 1 mM DTT). Vps4 constructs at monomer concentrations ranging from 0–60 µM were incubated with 3 nM of fluorescently labeled Vta1[VSL] in a total volume of 60 µl. Fluorescence polarization was read at equilibrium and apparent dissociation constants were estimated as described for the peptide binding studies (above), except that in this case Vps4 subunit concentrations were used for both graphing and fitting because each subunit contains a potential binding site for the VSL dimer. The binding model does not account for potential differences in affinity to different binding sites in the context of the asymmetric Vps4 hexamer, and $K_D$ values are therefore referred to as apparent dissociation constants.

## ATPase assay

The rate of ATP hydrolysis was determined using an end-point method modified from Merrill and Hanson (*Merrill and Hanson, 2010*). Vps4 at the indicated subunit concentration was incubated with 1 mM ATP at 37°C in 20 mM HEPES/NaOH pH 7.4, 100 mM NaCl, 10 mM MgCl$_2$, 1 mM DTT in a total volume of 10 µl. The hydrolysis reaction was stopped after 5 min by the addition of 100 µl of malachite green color reagent (14 mM ammonium molybdate, 1.3 M HCl, 1.5 mM malachite green) and 50 µl of 21% (w/v) citric acid. Absorbance at 650 nm was read using a Biotek Neo Synergy microplate reader and the amount of inorganic phosphate released by the reaction was determined using a sodium phosphate standard curve. Means and standard deviations reported in *Figure 1D* are from at least three independent experiments using different protein preparations with three or more technical replicates each.

## Analytical ultracentrifugation

To confirm that Vps4[101-437]-Hcp1 is hexameric in solution, we performed equilibrium sedimentation analyses at 4°C using an XLI analytical ultracentrifuge (Beckman Coulter, Indianapolis, IN) with absorbance optics. Sample cells with 6-channel centerpieces were filled with 120 µl of Vps4[101-437]-Hcp1 in 25 mM Tris/HCl pH 7.4, 100 mM NaCl at the indicated concentrations in the sample sectors and with 125 µl of buffer in the reference sectors. Absorbance scans at 280 nm were taken at equilibrium after centrifugation at 3000 rpm and 5000 rpm, respectively. Equilibrium sedimentation data were fit to a single species model in Heteroanalysis (*Cole, 2004*) using a theoretical molecular mass of 55,703.8 Da per subunit, a partial specific volume of 0.732662 mL/g, and a buffer density of 1.0049 g/mL, as calculated in SEDNTERP (*Hayes et al., 1995*).

## Glutaraldehyde crosslinking

For crosslinking with glutaraldehyde, proteins were buffer-exchanged by extensive dialysis in 20 mM HEPES/NaOH pH 7.4, 100 mM NaCl. Vps4[101-437]-Hcp1 (final subunit concentration 18 µM), Vta1[VSL] (final subunit concentration 36 µM) and the 8-residue peptide (from a 1 mM stock solution in water, final concentration 10 µM) were combined in the presence of 1 mM ADP·BeF$_x$ and 5 mM magnesium chloride in a total volume of 4.8 ml, and incubated on ice for 30 min before equilibration to room temperature over 5 min. Crosslinking was initiated by addition of 50 µl of 2% glutaraldehyde solution (diluted in dialysis buffer from an 8% stock, Fluka 49627, final concentration 0.02%), and quenched after 30 min by adding 5 ml of 1 M glycine containing 1 mM ADP·BeF$_x$ and 5 mM MgCl$_2$. Following concentration to 0.5 ml, glutaraldehyde and glycine were removed by gel filtration into 25 mM Tris/HCl pH 7.4, 100 mM NaCl, 1 mM ADP·BeF$_x$, 5 mM MgCl$_2$, and 1 mM DTT using a Superdex-200 column with a bed volume of 24 ml. The extent of crosslinking was assessed by SDS PAGE analysis of the peak fraction. The elution volume was as expected for a hexameric complex based on protein standards.

## Electron microscopy

3.5 µl of sample was applied to glow-discharged (25 mA, 25 s) Quantifoil 1.2/1.3 holey carbon 400 mesh copper grids, which were plunge frozen in liquid ethane using a Vitrobot Mark III (FEI,

Hillsboro, OR) set to 4°C, 80% relative humidity, 30 s wait time, −2 mm offset, and 8 s blotting time. Grids were stored in liquid nitrogen prior to data collection using SerialEM (*Mastronarde, 2005*) on a Tecnai TF20 (FEI) operating at 200 kV using a Gatan 626 side entry cryo-holder. Movies were recorded using a K2 Summit direct detector (Gatan, Pleasanton, CA) in counting mode at a corrected magnification of 70,952×, corresponding to a physical pixel size of 0.7047 Å, and at a dose rate of ~5 e⁻/pixel/sec. Each movie was recorded as a stack of 40 subframes, each of which was accumulated for 0.2 s, totaling ~80 electrons per Å². Defocus values ranged between 0.8 to 2.0 µm.

## Image processing and 3D reconstruction

Movie frames were aligned, exposure filtered, and summed into a single micrograph using Unblur (*Grant and Grigorieff, 2015*) (*Figure 2—figure supplement 1*). CTF parameters were determined using the program CTFFIND4 (*Rohou and Grigorieff, 2015*). Micrographs with poor CTF cross correlation scores were excluded from downstream analyses.

4059 particles were extracted from 41 micrographs after manual particle picking in EMAN2 using the *e2boxer.py* program (*Tang et al., 2007*) and used as input for non-CTF-corrected 2D class averaging in RELION (*Scheres, 2012*). The resulting 2D classes were used as templates for RELION autopicking, which resulted in extraction of 180,172 particles from 703 micrographs for full CTF-corrected image processing. After four rounds of 2D classification, 108,733 particles were identified as having Vps4-like features and used for an initial round of 3D classification (*Figure 2—figure supplement 1*). The initial model for templated Vps4 was generated using a gallery of low-pass filtered (40 Å) 2D classes in EMAN2 using the *e2initialmodel.py* program (*Tang et al., 2007*), which yielded a double-layered 3D structure that was consistent with the dimensions of Hcp1 and Vps4 (*Figure 2— figure supplement 2*). After 3D classification, 58,155 particles were identified as having ordered Vps4 features and used for RELION auto-refinement to generate an overall structure at 6.7 Å resolution (*Table 1*).

In order to optimize alignment on the Vps4 complex, we performed signal subtraction of Hcp1 densities in RELION using a previously described strategy (*Bai et al., 2015*). Briefly, a soft-edged mask for Hcp1 was generated by subtracting a soft-edged Vps4 mask from the soft-edged mask of the entire Hcp1-Vps4 complex (*Figure 2—figure supplement 2*). This Hcp1 mask was applied to the 6.7 Å resolution map calculated from the consensus refinement of 58,155 particles, and the resulting masked map was used for Hcp1 signal subtraction from raw particles based on the particle orientations determined from the consensus refinement. This generated a new stack of particle images and a new STAR file with updated metadata that was used as input for a new round of RELION 3D classification and auto-refinement, which resulted in a Vps4 map at 4.3 Å resolution calculated from 39,417 particles (*Table 1*, *Figure 2—figure supplements 1* and *2*). Local resolutions were estimated using ResMap (*Kucukelbir et al., 2014*) (*Figure 2—figure supplement 1*). Further quality control steps were taken by generating angular distribution plots, which confirmed a broad distribution of particle orientations, and comparisons between reference-free 2D class averages with 3D model reprojections of both the original consensus structure and the Hcp1-subtracted structure (*Figure 2— figure supplement 3*).

To exclude the possibility that crosslinking with 0.02% glutaraldehyde might stabilize an artificial conformation of the Vps4 hexamer, we collected and processed a data set of non-crosslinked Vps4^101-437-Hcp1 in complex with Vta1^VSL, ESCRT-III peptide and ADP·BeF_x. Samples were deposited on Quantifoil Graphene Oxide 2/4 200 mesh copper grids (SPI Supplies) glow-discharged for 25 s using a 10 mA current. Vitrification and data collection were performed as described above. 161,645 particles were extracted from 821 micrographs. After multiple rounds of 2D and 3D classification, particles were used for RELION auto-refinement, which yielded an ~13 Å resolution structure of the Vps4^101-437-Hcp1 particle. Both 2D and 3D classes showed Vps4 features similar to those seen with the glutaraldehyde-crosslinked sample (*Figure 2—figure supplement 4*). However, Vps4 features are much better defined when the structure is stabilized by crosslinking.

## Vta1 3D classification

Some weak density was observed at the expected site for Vta1^VSL at the Vps4 β domains (*Figure 2— figure supplement 6A*). We therefore performed focused 3D classification using the Hcp1-subtracted dataset to identify particles that contain the Vta1 density (*Figure 2—figure supplement 6B*).

This was performed separately around each Vps4 subunit $\beta$ domain, without particle re-alignment and by applying a generous soft-edged mask at the inter-subunit interface. The resulting classifications revealed Vta1 densities at each Vps4 subunit, and the corresponding particles were subjected to RELION auto-refinement. This strategy led to maps ranging between 5.3–7.2 Å resolution for the six sites (*Figure 2—figure supplement 6C*, *Figure 4—figure supplement 1* and *Table 2*). Focused 3D classification that encompassed multiple Vta1 regions failed to enrich for a single class containing multiple Vta1 densities, presumably because the occupancy of Vta1 sites is low in the vitrified sample.

## Vps4 subunit F 3D classification

The unsharpened Vps4 density map revealed reasonable density for subunit F at low contour levels (*Figure 2—figure supplement 7A*). We therefore performed focused 3D classification with a custom mask over subunit F to identify particles that contain ordered subunit F density (*Figure 2—figure supplement 7B*). The classification was performed without particle re-alignment (i.e., using the –skip_align flag in RELION) and revealed three distinct classes with ordered subunit F density. A fourth class containing 47% of particles showed poor subunit F density. Particles from the three classes with ordered density were used for separate RELION auto-refinement calculations, which led to maps ranging between 6.9–7.2 Å resolution (*Figure 2—figure supplement 7C-E, Table 3*). The maps were used for rigid body fitting of Vps4 coordinates into each subunit F position (*Figure 2—figure supplement 8*).

## Model building and refinement

Model building was facilitated by the availability of a Vps4 AAA ATPase cassette crystal structure (PDB 3EIE, [*Gonciarz et al., 2008*]). The AAA ATPase cassettes for subunits A-E were fit to the 4.3 Å map as rigid bodies and subjected to real-space refinement using Phenix (RRID:SCR_014224) (*Adams et al., 2010*) (*Figure 2—figure supplement 1F*). Secondary structure restraints were applied during refinement. Guided by visual inspection of map similarity, NCS restraints were applied to Vps4 subunits A-E with the exception of residues 240–247 and 260–267 of subunit A and residues 204–207 (pore loop 1) of subunit E. For Vps4 subunits A-E, residues 174–180 (P-loop) were restrained to a high resolution reference model (PDB 5BQ5, [*Arias-Palomo and Berger, 2015*]). For subunits A, B, and C, the distance between Be and the O3B of ADP was restrained to 1.6 Å, and the distance between Mg and F1 of BeF$_3$ was restrained to 2.0 Å. For subunits D and E the nucleotide was refined as ADP, while the subunit F nucleotide site was empty. Residues 204–207 (pore loop 1), 240–247 (pore loop 2) and 261–266 were absent in the previously reported structures and were built manually in Coot (RRID:SCR_014222) (*Emsley et al., 2010*). Because the 8-residue ESCRT-III peptide bound in the structure appears to occupy multiple sites, we did not attempt to build a detailed model but represented it as 8 Cα atoms in a low-energy extended conformation (*Figure 6*).

   To test for overfitting, all atoms in the refined model (of the AAA ATPase cassettes of subunits A-E and the peptide substrate) were randomly displaced by 0.5 Å and re-refined against one of the half maps derived from RELION auto-refinement. FSC curves for the re-refined model against the half map used for re-refinement (FSC$_{work}$) and against the other half map (FSC$_{test}$) showed close agreement (*Figure 2—figure supplement 1D*), consistent with lack of overfitting. The refined model was assessed using MolProbity (RRID:SCR_014226) (*Chen et al., 2010*) (*Table 1*).

   Models for Vta1$^{VSL}$ dimers and associated $\beta$ domains were built by rigid body docking of a previously reported structure (PDB 3MHV, [*Yang and Hurley, 2010*]). In cases where we observed

**Table 2.** Reconstruction statistics of Vps4-Vta1 classes.

| | Vps4 Vta1$^{VSL}$ (A-B) | Vps4 Vta1$^{VSL}$ (B-C) | Vps4 Vta1$^{VSL}$ (C-D) | Vps4 Vta1$^{VSL}$ (D-E) | Vps4 Vta1$^{VSL}$ (E-F) | Vps4 Vta1$^{VSL}$ (F-A) |
|---|---|---|---|---|---|---|
| Particle images | 26,243 | 13,066 | 10,700 | 11,684 | 14,274 | 26,964 |
| Resolution (unmasked, Å) | 6.9 | 7.8 | 7.8 | 7.8 | 7.5 | 6.9 |
| Resolution (masked, Å) | 5.4 | 6.7 | 7.2 | 6.9 | 6.5 | 5.3 |
| EMDB ID | EMD-8552 | EMD-8553 | EMD-8554 | EMD-8555 | EMD-8556 | EMD-8557 |

**Table 3.** Reconstruction statistics of Vps4-Subunit F classes.

| | Vps4 $F_1$ | Vps4 $F_2$ | Vps4 $F_3$ |
|---|---|---|---|
| Particle images | 6908 | 6794 | 7093 |
| Resolution (unmasked, Å) | 7.8 | 8.1 | 7.8 |
| Resolution (masked, Å) | 6.9 | 7.2 | 6.9 |
| EMDB ID | EMD-8572 | EMD-8571 | EMD-8570 |

additional density for N-terminal residues, the helix was extended accordingly. The $\beta$ domains of subunits A, B and C and the corresponding VSL domains were subjected to rigid body refinement, whereas other $\beta$ domains and VSL domains were positioned as docked by manual inspection. Subunit F was placed into the density by rigid-body fitting and not further refined. Finally, the model of the AAA ATPase cassettes for subunits A-E was combined with the models for Vta1$^{VSL}$ dimers and associated $\beta$ domains and subunit F. In order to obtain reasonable geometry, the connecting residues were regularized in Coot.

Figures of models and density maps were prepared using Chimera (RRID:SCR_004097) (*Pettersen et al., 2004*). Electrostatic potential was calculated using the Adaptive Poisson Boltzmann Solver (APBS, RRID:SCR_008387) (*Baker et al., 2001*) implemented in Chimera.

### Structure deposition

The complete model, including all 6 subunits of Vps4 AAA ATPase cassettes, 12 Vta1$^{VSL}$ domains, and the peptide, has been deposited into the PDB (RRID:SCR_012820) together with the unsharpened Hcp-masked map. The unmasked map (including both Hcp and Vps4), sharpened Hcp-masked map, and the 6 maps for the Vta1$^{VSL}$ domain were deposited at the EMDB (RRID:SCR_003207).

## Acknowledgements

Electron microscopy was performed at the University of Utah Electron Microscopy Core with support from David Belnap. We thank Anita Orendt and the Center for High Performance Computing at the University of Utah for computational support and resources. Sequencing was performed at the DNA Sequencing Core at the University of Utah, and mass spectrometry analysis was carried out by the Mass Spectrometry and Proteomics Core at the University of Utah, which obtained equipment through NCRR Shared Instrumentation Grant # 1 S10 RR020883-01 and 1 S10 RR025532-01A1. We thank James Fulcher and Michael Kay for peptide synthesis, Adam Nau for help with cloning, Frank Whitby for help with model refinement, and Janet Iwasa for making *Figure 8* and *Video 3*. This research was supported by NIH P50 GM082545 (W.I.S. and C.P.H.) and NIH T32 AI055434 (N.M.).

## Additional information

### Competing interests

WIS: Reviewing editor, *eLife*. The other authors declare that no competing interests exist.

### Funding

| Funder | Grant reference number | Author |
|---|---|---|
| National Institutes of Health | P50 GM082545 | Nicole Monroe<br>Han Han<br>Peter S Shen<br>Wesley I Sundquist<br>Christopher P Hill |
| National Institutes of Health | Microbial Pathogenesis Training Grant T32 AI055434 | Nicole Monroe |

The funders had no role in study design, data collection and interpretation, or the decision to submit the work for publication.

## Author contributions
NM, Formal analysis, Investigation, Visualization, Writing—review and editing; HH, Formal analysis, Validation, Investigation, Visualization; PSS, Data curation, Formal analysis, Validation, Investigation, Visualization, Writing—review and editing; WIS, Conceptualization, Resources, Supervision, Funding acquisition, Project administration; CPH, Conceptualization, Resources, Supervision, Funding acquisition, Validation, Writing—original draft, Project administration

## Author ORCIDs
Nicole Monroe, http://orcid.org/0000-0001-7678-4997
Han Han, http://orcid.org/0000-0003-0361-4254
Peter S Shen, http://orcid.org/0000-0002-6256-6910
Wesley I Sundquist, http://orcid.org/0000-0001-9988-6021
Christopher P Hill, http://orcid.org/0000-0001-6796-7740

# Additional files

## Supplementary files
• Supplementary file 1. Proteins and expression vectors.

## Major datasets
The following datasets were generated:

| Author(s) | Year | Dataset title | Dataset URL | Database, license, and accessibility information |
|---|---|---|---|---|
| Monroe N, Han H, Shen PS, Sundquist WI, Hill CP | 2017 | Vps4-Vta1 complex | http://www.rcsb.org/pdb/explore/explore.do?structureId=5UIE | Publicly available at the RCSB Protein Data Bank (accession no: 5UIE) |
| Monroe N, Han H, Shen PS, Sundquist WI, Hill CP | 2017 | Vps4-Vta1 complex | http://www.ebi.ac.uk/pdbe/entry/emdb/EMD-8549 | Publicly available at the EMBL-EBI Protein Data Bank (accession no: EMD-8549) |
| Monroe N, Han H, Shen PS, Sundquist WI, Hill CP | 2017 | Vps4-Vta1 complex_sharpened map | http://www.ebi.ac.uk/pdbe/entry/emdb/EMD-8550 | Publicly available at the EMBL-EBI Protein Data Bank (accession no: EMD-8550) |
| Monroe N, Han H, Shen PS, Sundquist WI, Hill CP | 2017 | Vps4-HCP hexamer | http://www.ebi.ac.uk/pdbe/entry/emdb/EMD-8551 | Publicly available at the EMBL-EBI Protein Data Bank (accession no: EMD-8551) |
| Monroe N, Han H, Shen PS, Sundquist WI, Hill CP | 2017 | Vps4-Vta1 complex_VSL_A | http://www.ebi.ac.uk/pdbe/entry/emdb/EMD-8552 | Publicly available at the EMBL-EBI Protein Data Bank (accession no: EMD-8552) |
| Monroe N, Han H, Shen PS, Sundquist WI, Hill CP | 2017 | Vps4-Vta1 complex_VSL_B | http://www.ebi.ac.uk/pdbe/entry/emdb/EMD-8553 | Publicly available at the EMBL-EBI Protein Data Bank (accession no: EMD-8553) |
| Monroe N, Han H, Shen PS, Sundquist WI, Hill CP | 2017 | Vps4-Vta1 complex_VSL_C | http://www.ebi.ac.uk/pdbe/entry/emdb/EMD-8554 | Publicly available at the EMBL-EBI Protein Data Bank (accession no: EMD-8554) |
| Monroe N, Han H, Shen PS, Sundquist WI, Hill CP | 2017 | Vps4-Vta1 complex_VSL_D | http://www.ebi.ac.uk/pdbe/entry/emdb/EMD-8555 | Publicly available at the EMBL-EBI Protein Data Bank (accession no: EMD-8555) |
| Monroe N, Han H, Shen PS, Sundquist | 2017 | Vps4-Vta1 complex_VSL_E | http://www.ebi.ac.uk/pdbe/entry/emdb/EMD- | Publicly available at the EMBL-EBI Protein |

| | | | | | |
|---|---|---|---|---|---|
| WI, Hill CP | | | | 8556 | Data Bank (accession no: EMD-8556) |
| Monroe N, Han H, Shen PS, Sundquist WI, Hill CP | 2017 | Vps4-Vta1 complex_VSL_F | | http://www.ebi.ac.uk/pdbe/entry/emdb/EMD-8557 | Publicly available at the EMBL-EBI Protein Data Bank (accession no: EMD-8557) |
| Monroe N, Han H, Shen PS, Sundquist WI, Hill CP | 2017 | Vps4-Vta1 complex, State 3 of subunitF | | http://www.ebi.ac.uk/pdbe/entry/emdb/EMD-8570 | Publicly available at the EMBL-EBI Protein Data Bank (accession no: EMD-8570) |
| Monroe N, Han H, Shen PS, Sundquist WI, Hill CP | 2017 | Vps4-Vta1 complex, State 2 of subunitF | | http://www.ebi.ac.uk/pdbe/entry/emdb/EMD-8571 | Publicly available at the EMBL-EBI Protein Data Bank (accession no: EMD-8571) |
| Monroe N, Han H, Shen PS, Sundquist WI, Hill CP | 2017 | Vps4-Vta1 complex, State 1 of subunitF | | http://www.ebi.ac.uk/pdbe/entry/emdb/EMD-8572 | Publicly available at the EMBL-EBI Protein Data Bank (accession no: EMD-8572) |

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
