## [Decision Letter]

Thank you for submitting your article "Structural basis of protein translocation by the Vps4-Vta1 AAA ATPase" for consideration by *eLife*. Your article has been reviewed by three peer reviewers, and the evaluation has been overseen by Sriram Subramaniam as the Reviewing Editor and Michael Marletta as the Senior Editor. The following individuals involved in review of your submission have agreed to reveal their identity: James M Berger and Janet Vonck.

The reviewers have discussed the reviews with one another and the Reviewing Editor has drafted this decision to help you prepare a revised submission.

Summary:

The manuscript by Monroe et al. details the structure of the ESCRT-III disassembly protein, Vps4, in complex with the activating portion of its co-factor, Vta1, and both nucleotide and a peptide substrate. Vps4, a hexameric AAA+ ATPase, is found to adopt an asymmetric configuration in which five of its subunits adopt a spiral staircase configuration around what turns out to be an extended client peptide configuration; the sixth subunit is found to adopt several different positions between the upper and lower ends of the staircase. Vta1 is seen to bind to the edge of the Vps4 hexamer, in some instances serving as a bridge between adjoining subunits. The EM data acquisition and image analysis methodology is classical and seems solid. 3D classification and focused classification in RELION are used to resolve mobile or substoichiometric features.

The paper is well organized and an easy, enjoyable read. The structural data are used to suggest that this ATPAse unfolds client proteins by a rotary translocation mechanism, a model that seems eminently sensible based on the states captured here. The model of the Vps4 helix rotating along the unfolded ESCRT-III substrate in an ATP-dependent fashion is appealing and stimulating. The figures are mostly clear, but in some cases, it is difficult to follow where certain structural elements are located relative to each other, as detailed below.

Essential revisions:

1) Protein stabilization and preparation

The authors make the assumption that the Hcp1 protein fused to Vps4 to promote assembly does not influence the structure they observe. Since Hcp1 is a planar, C6 hexamer, and Vps4 is clearly asymmetric, this assumption seems reasonable. However, the authors should comment on this point a little more. For instance, is the high level of ATPase activity seen for Hcp1-fused Vps4 comparable to that of the wild-type protein on a native ESCRT- III assembly? Does changing the linker length between the Vsp4 and Hcp1 portions of the fusion protein alter activity? Additionally, regarding the fact that the Vps4 hexamer is stabilized by fusion to the hexameric Hcp1 and even further by 0.02% glutaraldehyde: although the ATPase activity is retained (Figure 1), could the authors please address the possibility that this artificial stabilization by cross-linking may have implications for the physiological relevance of the structures determined in this manuscript. Finally, please provide more information regarding concentrations of proteins, buffer, and the reconstitution protocol.

2) Specific comments on 3D reconstruction to be addressed

A) The signal of Hcp1 was subtracted from the original images as proposed by Bai et al., 2015 to generate a new stack of images used for further refinement of the Vps4-Vta1^VSL^-ESCRT-III part. Why didn't the authors then return to the original, non-subtracted particles for further 3D auto-refinement runs as a control as described in Bai et al.?

B) How and which NCS restraints are applied during atomic model refinement considering that the pseudo-helical complex is asymmetric?

C) It would be valuable to show angular distribution plots and a comparison of 3D model re-projections to several 2D class averages.

D) Subsection “Model Building and Refinement”: "features absent in the previously reported structures were built manually in Coot." Considering the resolution of 4.3 Å, this is not trivial. Please detail which features are involved.

3) Remarks on the text and figures

A) Should BeF_x_ be noted as BeF_3_ throughout the paper? One of the reviewers notes that he/she is unaware of beryllium adopting anything other than a tri-fluoride liganded state, unlike aluminum, which can bind three or four fluorine atoms depending on solution pH. If the authors know this assumption to be false, please provide a few references.

B) Subsection “Comparison with other AAA ATPases”. In what way are the hexameric structures of E1 and Rho distinct from Vps4? It seems that the model proposed here is actually rather similar to their proposed mechanisms of translocation.

C) Discussion – how does the state of the ATPase region and mechanism compare to NSF? It would be useful to build upon the proteasome comparison with some additional examples, bringing up E1/Rho here as well.

D) Figure 1: Do the concentration values refer to peptide concentration?

E) Figure 1 does not really show that the resulting complex is indeed hexameric, a combination of Size Exclusion Chromatography with Multi-Angle Light Scattering analysis would be more appropriate.

F) Figure 2/Figure 2—figure supplement 7; Figure 2—figure supplement 8: It appears that the densities for the subunit F shown in Figure 2 and Figure 2—figure supplement 7 and Figure 2—figure supplement 8 don't look like what would be expected for 6.9-7.2 A maps, the α-helices are not clearly resolved. Could it be that the resolution is inflated by the mask used for focus classification?

G) How different are F1, F2 and F3 from each other? Any details?

H) Figure 2: Introduction, second paragraph: a reference to Figure 2 would be very helpful to understanding of the structure.

I) Figure 2—figure supplement 1 refers to a nonexistent "Supplementary Figure 3". Please give the correct figure number.

J) Figure 6: Subsection “ESCRT-III substrate peptide binds close to the helix axis of the central pore”: the location of the L1 loop and L2 loop in the structure is not clear. This could be indicated in Figure 2. In Figure 6 and Figure 7—figure supplement 1, the same feature is called "pore loop", which is presumably an interpretation (loops in the central pore of the hexamer?).

K) Figure 6: please indicate the color code and the amino acid numbers (is the pore loop the same as L1 loop?).

L) Figure 7 is too small.

M) Subsection “Vta1 dimers bind two adjacent Vps4 subunits”, second paragraph: "that β domain" should be "the β domain".

N) Subsection “Vta1 dimers bind two adjacent Vps4 subunits”, last paragraph and Figure 5: the location of Vps4 L151 is not indicated.

---

## [Author Response]

*Essential revisions:*

*1) Protein stabilization and preparation*

*The authors make the assumption that the Hcp1 protein fused to Vps4 to promote assembly does not influence the structure they observe. Since Hcp1 is a planar, C6 hexamer, and Vps4 is clearly asymmetric, this assumption seems reasonable. However, the authors should comment on this point a little more. For instance, is the high level of ATPase activity seen for Hcp1-fused Vps4 comparable to that of the wild-type protein on a native ESCRT- III assembly? Does changing the linker length between the Vsp4 and Hcp1 portions of the fusion protein alter activity? Additionally, regarding the fact that the Vps4 hexamer is stabilized by fusion to the hexameric Hcp1 and even further by 0.02% glutaraldehyde: although the ATPase activity is retained (Figure 1), could the authors please address the possibility that this artificial stabilization by cross-linking may have implications for the physiological relevance of the structures determined in this manuscript.*

There are several observations that give confidence that the Hcp1 fusion and glutaraldehyde crosslinking do not distort the Vps4 structure:

(i) The revised manuscript now includes new data from analytical ultracentrifugation that demonstrate that the Vps4-Hcp1 fusion protein is hexameric. This is consistent with our earlier published finding that authentic Vps4 is active as a hexamer (Monroe et al., J.Mol.Biol. 426:510-25, 2014).

(ii) As the reviewers note, Hcp1 has not imposed its strict 6-fold rotational symmetry on Vps4, which displays marked asymmetry.

(iii) The fusion protein displays the same affinity for substrate ESCRT-III peptide as the corresponding Vps4 construct lacking the fusion partner. See Figure 1AB of the revised manuscript.

(iv) The fusion protein displays ~5-fold higher ATPase activity compared to the corresponding Vps4 construct lacking the fusion partner. These assays are done at relatively low Vps4 concentration where most of the Vps4 is disassembled, and indicate that Hcp1 is promoting formation of the active Vps4 hexamer. See Figure 1 of the revised manuscript.

(v) The closest approach of Vps4 and Hcp1 atoms that have been modeled in the overall consensus structure is 7Å. This distance is too long for a direct Vps4-Hcp1 interaction and, moreover, those atoms are at the ends of inherently flexible side chains. Moreover, the Vps4 C-termini and the corresponding Hcp1 N-termini are separated by 21-31Å, which can easily be spanned by the 18-residue Vps4-Hcp1 linker, which in a fully extended conformation would be longer than 60Å. This point is noted in the first paragraph of the Results and Discussion section of the revised manuscript.

(vi) To address the concern that glutaraldehyde crosslinking might be altered the structure, we have determined the structure in the absence of glutaraldehyde crosslinking. This structure only extends to an overall resolution of 10Å, and the Vps4 portion is more poorly resolved than that. Nevertheless, the resulting low resolution structure clearly resembles the higher resolution structure determined for the glutaraldehyde-crosslinked fusion protein. These new data are shown in the revised manuscript as (Figure 2—figure supplement 4).

*Finally, please provide more information regarding concentrations of proteins, buffer, and the reconstitution protocol.*

This has been included in the Methods section of the revised manuscript.

*2) Specific comments on 3D reconstruction to be addressed*

*A) The signal of Hcp1 was subtracted from the original images as proposed by Bai et al., 2015 to generate a new stack of images used for further refinement of the Vps4-Vta1^VSL^-ESCRT-III part. Why didn't the authors then return to the original, non-subtracted particles for further 3D auto-refinement runs as a control as described in Bai et al.?*

The motivation for signal subtraction is different between our paper and Bai et al. in two important respects:

(i) Bai et al. used signal subtraction to improve their focused 3D classification of a small (~30kDa) component of their complex. Thus, their control of returning to non-subtracted particles was needed to confirm that the focused classification indicated a component that was indeed part of the entire complex. In contrast, our goal was to improve the alignment over a large region (Vps4) that was already clearly visible before applying Hcp1 signal subtraction.

(ii) Unlike the case with Bai et al., the subtracted region in our study (Hcp1) is not really part of the functional Vps4 complex, but is just attached by an artificial flexible tether. Thus, in our case, returning to non-subtracted particles would yield lower resolution for a structure that does not need this validation and is not functionally relevant (Hcp1).

*B) How and which NCS restraints are applied during atomic model refinement considering that the pseudo-helical complex is asymmetric?*

We have added the following information to the Methods section “Model Building and Refinement” of the revised manuscript:

“Guided by visual inspection of map similarity, NCS restraints were applied to Vps4 subunits A-E with the exception of residues 240-247 and 260-267 of subunit A and residues 204-207 (pore loop 1) of subunit E.” This method is appropriate for an asymmetric structure such as the Vps4 hexamer because it makes no assumption of point group symmetry, and follows the standard approach of applying NCS restraints in the refinement of crystal structures where multiple copies of a molecule are found in the asymmetric unit in the absence of point group symmetry.

C) It would be valuable to show angular distribution plots and a comparison of 3D model re-projections to several 2D class averages.

As requested, the revised manuscript includes a new figure (Figure 2—figure supplement 3) that shows angular distribution plots and comparisons between reference-free 2D class averages and 3D model re-projections.

*D) Subsection “Model Building and Refinement”: "features absent in the previously reported structures were built manually in Coot." Considering the resolution of 4.3 Å, this is not trivial. Please detail which features are involved.*

The revised manuscript now explicitly lists the features that were absent in previous models and that we have built manually in the Methods section “Model Building and Refinement”. Specifically, “Residues 204-207 (pore loop 1), 240-247 (pore loop 2) and 261-266 were absent in the previously reported structures and were built manually in Coot (Emsley et al., 2010).”

Despite the modest resolution, a number of factors facilitated the building of reasonable models in these regions: (i) The ends of these missing loops are constrained by the previously reported crystal structures. (ii) The missing loops are short. (iii) The presence of large side chains such as W206 and M207 in pore loop 1 served as helpful guides. (iv) The density for these regions was closely similar between the multiple Vps4 subunits in the complex.

*3) Remarks on the text and figures*

*A) Should BeF_x_ be noted as BeF_3_ throughout the paper? One of the reviewers notes that he/she is unaware of beryllium adopting anything other than a tri-fluoride liganded state, unlike aluminum, which can bind three or four fluorine atoms depending on solution pH. If the authors know this assumption to be false, please provide a few references.*

Beryllium is reported to form a series of complexes with fluoride that are all tetragonal and coordinate 1-4 fluoride ions, with the other positions being occupied by water or hydroxyl anion (Mesmer RE & Baes CF, Jr, Inorg. Chem. 8, 618-626, 1967). At pH 7.5 and at a low millimolar fluoride concentration, as used in this study, the species acting as a phosphate analog is expected to be a mixture of BeF_2_(OH)^-^·H_2_O and BeF_3_^-^·H_2_O (Combeau & Carlier, J. Biol. Chem. 264 19017-19021, 1989, and references therein). We therefore prefer to denote beryllium fluoride as BeF_x_.

*B) Subsection “Comparison with other AAA ATPases”. In what way are the hexameric structures of E1 and Rho distinct from Vps4? It seems that the model proposed here is actually rather similar to their proposed mechanisms of translocation.*

Our proposed mechanism for Vps4 is conceptually very similar to those of E1 and Rho in that sequential ATP binding and hydrolysis around the hexameric ring drives propagation of a helical set of subunits that present a binding site for the substrate. The revised manuscript explicitly states this in the “Comparison with other AAA ATPases” segment of the Results and Discussion section by inclusion of the paragraph: “Structures of the substrate-bound DNA helicase E1 (Enemark and Joshua-Tor, 2006) and RNA translocase Rho (Thomsen and Berger, 2009, Thomsen et al., 2016) suggested sequential mechanisms of translocation that are conceptually analogous to our proposal for translocation of protein by Vps4. Thus, the structures indicate that Vps4 and the hexameric nucleic acid translocases function by forming a helical arrangement of subunits that matches the symmetry of their translocating substrate, with one or two transitioning subunits that are disengaged from the substrate, and with translocation achieved by sequential propagation of the ATPase helix.” A more detailed comparison of the structures is interesting but beyond the scope of the current manuscript.

*C) Discussion – how does the state of the ATPase region and mechanism compare to NSF? It would be useful to build upon the proteasome comparison with some additional examples, bringing up E1/Rho here as well.*

We agree that this is a very interesting topic, and is one that we are exploring. It is summarized in the final paragraph of the Results and Discussion section: “The structure of Vps4 superimposes with some other hexameric AAA ATPases, including the ATPases of the 26S proteasome, which, like Vps4, translocate protein substrates and adopt a right-handed helical notched-washer structure (Forster et al., 2013, Lander et al., 2013). […] It will be of considerable interest to determine the extent to which the geometry of substrate binding and the cycles of ATP-induced helix propagation envisioned here in light of the Vps4-substrate complex underlie the mechanisms of other protein-translocating AAA ATPases, and the extent to which variations on the idealized model, such as by variations in peptide binding geometry or in the sequence and timing of ATP hydrolysis, may apply.” A more detailed comparison of the structures is interesting but beyond the scope of the current manuscript.

*D) Figure 1: Do the concentration values refer to peptide concentration?*

Concentration values refer to Vps4 subunit concentrations. The revised manuscript states this information explicitly in the figure legend and in the Methods section.

*E) Figure 1 does not really show that the resulting complex is indeed hexameric, a combination of Size Exclusion Chromatography with Multi-Angle Light Scattering analysis would be more appropriate.*

We agree that the hexameric state of the Vps4 complex was not demonstrated unambiguously by SEC. The revised manuscript now resolves this concern by including new data from analytical ultracentrifugation that shows that Vps4-Hcp1:Vta1:ESCRT-III is indeed hexameric in solution. See Figure 1—figure supplement 1 of the revised manuscript.

*F) Figure 2/Figure 2—figure supplement 7; Figure 2—figure supplement 8: It appears that the densities for the subunit F shown in Figure 2 and Figure 2—figure supplement 7; Figure 2—figure supplement 8 don't look like what would be expected for 6.9-7.2 A maps, the α-helices are not clearly resolved. Could it be that the resolution is inflated by the mask used for focus classification?*

The overall resolution of these maps is 6.9-7.2 Å, but the local resolution over subunit F is poorer than that. Indeed, individual helices are readily resolved for subunits A-E of these maps. The overall position of subunit F is clear, but not defined at the level of visualizing distinct helices in the maps. We have made this point explicit by the inclusion of local resolution maps in Figure 2—figure supplement 7.

*G) How different are F1, F2 and F3 from each other? Any details?*

This is shown in the revised manuscript in Figure 2—figure supplement 8 and in Figure 7.

*H) Figure 2: Introduction, second paragraph: a reference to Figure 2 would be very helpful to understanding of the structure.*

We inserted a reference to Figure 2 in this paragraph.

*I) Figure 2—figure supplement 1 refers to a nonexistent "Supplementary Figure 3". Please give the correct figure number.*

Thank you. We have changed the figure call to Figure 2—figure supplement 6.

*J) Figure 6: Subsection “ESCRT-III substrate peptide binds close to the helix axis of the central pore”: the location of the L1 loop and L2 loop in the structure is not clear. This could be indicated in Figure 2. In Figure 6 and Figure 7—figure supplement 1, the same feature is called "pore loop", which is presumably an interpretation (loops in the central pore of the hexamer?).*

To be more consistent with prior publications, L1 and L2 have been renamed pore loop 1 and pore loop 2 throughout the revised manuscript. These features are now labeled explicitly in Figure 2 and are highlighted in an insert to that figure.

*K) Figure 6: please indicate the color code and the amino acid numbers (is the pore loop the same as L1 loop?).*

The revised manuscript includes this information in the legend for Figure 6, which now reads “Stereo view of the peptide and pore loop 1 (residues 203-210) of subunits A-D with density.”.

*L) Figure 7 is too small.*

The size of Figure 7 has been increased in the revised manuscript.

M) Subsection “Vta1 dimers bind two adjacent Vps4 subunits”, second paragraph: "that β domain" should be "the β domain".

Thank you. This has been corrected in the revised manuscript.

*N) Subsection “Vta1 dimers bind two adjacent Vps4 subunits”, last paragraph and Figure 5: the location of Vps4 L151 is not indicated.*

Vps4 L151 is located in the hexamer interface. The revised manuscript indicates the location of this residue in Figure 2.